



# Aerosol spectral optical properties in the Paris urban area, and its peri−urban and forested surroundings during summer 2022 from ACROSS surface observations

Ludovico Di Antonio[1], Claudia Di Biagio[2], Paola Formenti[2], Aline Gratien[2], Vincent Michoud[2], Christopher Cantrell[1], Astrid Bauville[1], Antonin Bergé[1,a], Mathieu Cazaunau[2], Servanne Chevaillier[2], Manuela Cirtog[1], Patrice Coll[2], Barbara D'Anna[3], Joel F. de Brito[4], David O. De Haan[5], Juliette R. Dignum[5], Shravan Deshmukh[6], Olivier Favez[7,8], Pierre-Marie Flaud[9], Cecile Gaimoz[1], Lelia N. Hawkins[10], Julien Kammer[3], Brigitte Language[1,b], Franck Maisonneuve[1], Griša Močnik[11], Emilie Perraudin[9], Jean-Eudes Petit[12], Prodip Acharja[12], Laurent Poulain[6], Pauline Pouyes[9], Eva Drew Pronovost[10], Véronique Riffault[4], Kanuri I. Roundtree[5], Marwa Shahin[3], Guillaume Siour[1], Eric Villenave[9], Pascal Zapf[2], Gilles Foret[1], Jean-François Doussin[1], and Matthias Beekmann[2]

[1]Univ Paris Est Creteil and Université Paris Cité, CNRS, LISA, F−94010 Créteil, France
[2]Université Paris Cité and Univ Paris Est Creteil, CNRS, LISA, F−75013 Paris, France
[3]Aix Marseille Univ, CNRS, LCE, Marseille, France
[4]Centre for Energy and Environment, IMT Nord Europe, Institut Mines−Télécom, Université de Lille, Lille, 59000, France
[5]Department of Chemistry and Biochemistry, University of San Diego, 5998 Alcala Park, San Diego, California 92110, United States of America
[6]Leibniz Institute for Tropospheric Research (TROPOS), Leipzig 04318, Germany
[7]Institut national de l'environnement industriel et des risques (Ineris), Parc Technologique Alata BP2, 60550 Verneuil−en−Halatte, France
[8]Laboratoire Central de Surveillance de la Qualité de l'Air (LCSQA), 60550 Verneuil−en−Halatte, France
[9]Univ. Bordeaux, CNRS, EPOC, EPHE, UMR 5805, F-33600 Pessac, France
[10]Department of Chemistry, Harvey Mudd College, 301 Platt Blvd, Claremont, California 91711, United States
[11]Center for Atmospheric Research, University of Nova Gorica, Nova Gorica SI−5000, Slovenia
[12]Laboratoire des Sciences du Climat et de l'Environnement, CEA−CNRS−UVSQ, IPSL, Université Paris−Saclay, 91191 Gif−sur−Yvette, France
[a]now at Laboratoire des Sciences du Climat et de l'Environnement, CEA−CNRS−UVSQ, IPSL, Université Paris−Saclay, 91191 Gif−sur−Yvette, France
[b]now at Unit for Environmental Sciences and Management, North−West University, Potchefstroom, South Africa

*Correspondence to*: Ludovico Di Antonio (ludovico.diantonio@lisa.ipsl.fr), Claudia Di Biagio (claudia.dibiagio@lisa.ipsl.fr)

**Abstract.** The complex refractive index (CRI, n−ik) and the single scattering albedo (SSA) are key parameters driving the aerosol direct radiative effect. Their spatial, temporal, and spectral variabilities in anthropogenic−biogenic mixed environments are poorly understood. In this study, we retrieve the spectral CRI and SSA (370−950 nm wavelength range) from *in situ* surface optical measurements and number size distribution of submicron aerosols at three sites in the greater Paris area, representative of the urban city, its peri−urban and forested rural environments. Measurements were taken as part of the ACROSS (Atmospheric ChemistRy Of the Suburban forest) campaign in June−July 2022 under diversified conditions: 1) two heatwaves leading to high aerosol levels; 2) an intermediate period with low aerosol concentrations; 3) an episode of long−range transported fire emissions. The retrieved CRI and SSA exhibit an urban–to–rural gradient, whose intensity is modulated by the weather conditions. A full campaign average CRI of 1.41-0.037i (urban), 1.52-0.038i (peri−urban), 1.50−0.025i (rural) is retrieved. The imaginary part of the CRI (k) increases and the SSA decreases at the peri−urban and forest sites when exposed to the influence of the Paris urban plume. Values of k>0.1 and SSA<0.6 at 520 nm are related to black carbon mass fraction larger than 10%. Organic aerosols are found to contribute to more than 50% of the aerosol mass and up to 10% (urban), 17% (peri−urban) and 22% (forest) of the aerosol absorption coefficient at 370 nm. A k of 0.022 (370 nm) was measured at the urban site for the long−range transported fire episode.



## 1. Introduction

Climate change represents one of the most serious challenges that society is facing today. The scientific community has demonstrated that human activities are leading to a global temperature increase, impacting the weather and climate over different regions of the world (IPCC, 2022). Aerosol particles, both from natural and anthropogenic sources, play a major role in the climate system (Boucher et al., 2013; Forster et al., 2021; Szopa et al., 2021). Depending on their size distribution, morphology, mixing state, and chemical composition, they affect the radiative budget of the Earth−atmosphere system both directly, by scattering and absorbing solar and thermal radiations (the aerosol−radiation interaction effect, ARI), and indirectly by affecting the surface albedo and the microphysical properties and lifetime of clouds (the aerosol−cloud interaction effect, ACI) (Bellouin et al., 2020; Boucher et al., 2013). While tropospheric anthropogenic aerosols are considered as the second most important contributors to the global and regional radiative forcing, the estimation of the magnitude and sign of this effect still remains uncertain (IPCC, 2021). In fact, due to the high spatial and temporal variability in aerosol sources, distribution and properties, the representation of aerosols in models remains a challenge (Bellouin et al., 2020; Bender, 2020; Li et al., 2022).

The magnitude and sign of the aerosol direct radiative effect is determined by the particle spectral optical properties, represented by the complex refractive index (CRI) and the single scattering albedo (SSA) (Samset et al., 2018; Zhou et al., 2018). The CRI ($n(\lambda)-ik(\lambda)$) represents the particle ability to scatter and absorb solar radiation. Its real part ($n$) is related to the aerosol non−absorbing component, while its imaginary part ($k$) is associated to the absorbing one (Bohren et Huffman, 1983; Seinfeld and Pandis, 2016). Most of the aerosol inorganic species (e.g. sulfate and nitrate) show a very weak CRI imaginary part and a SSA close to 1 in the solar spectrum (Aouizerats et al., 2010; Chang et al., 2022; Freedman et al., 2009; Mao et al., 2022; Moore et al., 2021). Aerosols containing black carbon (BC) and brown carbon (BrC; i.e. the absorbing fraction of organic particles) absorb a significant fraction of incoming solar radiation and may cause local atmospheric warming, inducing the evaporation of cloud droplets and the reduction of the atmospheric stability (the semi−direct effect, (Allen et al., 2019)). Nevertheless, the contribution of BC− and BrC−containing aerosols to the radiative budget at both regional and global scales in models features among the highest uncertainties in the climate forcing assessment (Kelesidis et al., 2022; Räisänen et al., 2022; Sand et al., 2021; Tuccella et al., 2021, 2020; Bond et al., 2013).

BC is emitted predominantly by combustion processes associated with traffic, industrial activities and wildfires (Bond et al., 2013). One of the most widely used CRI in climate models for BC is 1.95−0.79i (wavelength independent), showing a strong imaginary component and resulting in a SSA of around 0.3 (at 550 nm) (Bond and Bergstrom, 2006; Samset et al., 2018). However, BC may exist under different mixing states: i) external mixing (e.g. freshly emitted BC) and ii) internal mixing with other aerosol species (e.g. aged BC), generally inducing a (wavelength−dependent) light absorption enhancement (Hu et al., 2022; Kalbermatter et al., 2022; Liu et al., 2017; Saleh et al., 2015; Bond et al., 2006; Jacobson, 2001).

While for a long time organic carbon (OC) has been assumed to be non−absorbing in models, OC is now recognized to play an important role in the radiative forcing (Li et al., 2023; Ferrero et al., 2021; Saleh et al., 2015; Tuccella et al., 2020, 2021; Wang et al., 2018; Zhang et al., 2020a). The BrC absorbs light, mainly in the near−ultraviolet (UV) part of the light spectrum (Kirchstetter et al., 2004). It is emitted primary by fossil fuel, biofuel and biomass burning combustion or generated as a secondary species resulting from gas and aqueous phase reactions, including oxidation of anthropogenic and biogenic volatile organic compounds (VOCs), leading to absorbing biogenic− and anthropogenic− secondary organic aerosols (BSOAs and ASOAs) (Betz et al., 2022; He et al., 2021; Laskin et al., 2015; Liu et al., 2016, 2015; Moise et al., 2015; Updyke et al., 2012; Xiong et al., 2022). The absorbing capacity of BrC, and therefore its spectral CRI and SSA, might vary along with the process of formation, the precursor type, the oxidation patterns and the $NO_x$ levels for secondary species (Dingle et al., 2019; Flores et al., 2014; He et al., 2021; Kim and Paulson, 2013; Liu et al., 2015; Moise et al., 2015; Nakayama et al., 2012, 2015; Yang et al., 2022), as well as due to chemical ageing, potentially inducing either bleaching or browning of the brown aerosols



(Velazquez-Garcia et al., 2024; Zhang et al., 2020a; Wang et al., 2018; Liu et al., 2015; Moise et al., 2015). The biogenic−derived secondary organic aerosols have a weaker CRI imaginary component compared to the highly absorbing SOA from anthropogenic VOCs oxidization at high $NO_x$ levels (He et al., 2021; Liu et al., 2015). Finally, laboratory and field

investigations suggest that the mixing of anthropogenic and biogenic precursors under several $NO_x$ levels could impact the absorption properties of the SOA (He et al., 2022; Hecobian et al., 2010; Liu et al., 2016; Moise et al., 2015).

While aerosol composition and properties have been studied extensively in urban polluted environments (Boedicker et al., 2023; Cappa et al., 2019; Che et al., 2009; Ebert et al., 2004; Hu et al., 2016; Kahnert and Kanngießer, 2020; Kirago et al., 2022; Nault et al., 2021; Xie et al., 2019), the regional scale evolution and fate of those urban polluted air masses have been

addressed in only few studies such as the recent GO−Amazon or KORUS campaigns in the Amazon basin and South Korea (Crawford et al., 2021; LeBlanc et al., 2022; Nascimento et al., 2021; Shrivastava et al., 2019). The outflow of urban pollution towards peri−urban and rural environments leads in fact to the mixing of primary and secondary anthropogenic and biogenic compounds and varying BC, VOCs, and $NO_x$ levels, in turn influencing the aerosol production, ageing, composition, mixing state and climate−relevant properties. The lack of observations on such evolving conditions hampers the proper representation

of aerosols in chemical−transport and regional climate models, and our capacity to understand and predict the current and future evolution of climate and its feedbacks.

In this paper we investigate the spectral aerosol CRI and SSA (from 370 to 950 nm wavelengths) and its regional gradient from urban to rural environment using *in situ* surface measurements of optical properties (absorption and scattering coefficients) and number size distribution of submicron particles at three ground−based sites in the Paris area (Ile−de−France

region, see Fig. 1). Measurements were conducted in June−July 2022 as part of the ACROSS (Atmospheric ChemistRy Of the Suburban forest) project (Cantrell and Michoud, 2022). The Paris metropolitan area targeted by ACROSS is the most populated urban area in France (around 12 million inhabitants, https://www.insee.fr/), located more than 200 km from other major agglomerations, and surrounded by forested and agricultural areas. In order to measure the diversity of emissions and the evolution of air mass properties in the region, measurements were deployed at three ground−based sites: i) a site in central

urban Paris (Paris Rive−Gauche, PRG), ii) the SIRTA (the Site Instrumental de Recherche par Télédétection Atmosphérique, located 15 km south west of Paris) facility representative of peri−urban conditions, and iii) a site in the Rambouillet forested area (RambForest, supersite of the project, located 50 km south west downwind of Paris). The ACROSS campaign took place between 14 June to 25 July 2022, a period characterized by extreme weather conditions and high temperatures for Europe (Copernicus Climate Change Service report;(C3S, 2023)). Various situations were sampled during ACROSS, including: 1)

two strong heatwaves, promoting the biogenic SOA build−up and wind conditions favoring the export of the Paris emission plume towards the peri−urban and forest sites; 2) a period between the two heatwaves, characterized by low aerosol concentrations and limited urban outflow. Additionally, two concurrent situations occurred during the campaign: 3) an intense biomass burning episode rising up a smoke plume from southern France (i.e. the Landes forest) to the Ile−de−France area; 4) two Saharan dust events advecting coarse−sized dust aerosols from the upper atmosphere to the ground.

This paper is organized as follows: the ground−based sites, instrumentation, and measurements are presented in Section 2. The CRI and SSA retrieval procedures are detailed in Section 3. The aerosol spectral absorption coefficient, scattering coefficient, CRI and SSA variability within the different ACROSS periods, as well as the link to meteorology and aerosol bulk chemical composition, are presented and discussed in Sections 4 and 5. Conclusions are drawn in Section 6.



## 2. Methods

### 2.1 Site description

The ACROSS campaign included ground−based and aircraft observations across the Ile−de−France region deploying a large panel of instruments for measuring both the gas and the aerosol phases (Cantrell and Michoud, 2022). The three ground−based sites considered in this study (Fig. 1) are briefly described in the following.

The Paris–Rive Gauche (PRG) site (48.8277° N, 2.3806 °E; hereafter PRG) was hosted at the Lamarck building at Université
Paris Cité, in the southeastern part of the Paris administrative borders. Within the Paris urban agglomeration and in close proximity to the urban area in the east, this site presents urban background features: it is at a distance of a few hundred meters from strong emission sources such as bus and train stations, main roads and city highway intersections. Aerosol and gas sampling was performed from the roof of the building at about 30 m a.g.l.. A complete description of the site deployment during ACROSS, including gas and aerosol measurements will be provided separately (Di Biagio et al., in preparation).

The SIRTA (Site Instrumental de Recherche par Télédétection Atmosphérique, 48.7090° N, 2.1488° E) is an Aerosol, Clouds, and Trace gases Research Infrastructure (ACTRIS) long−term observational facility (Bedoya-Velásquez et al., 2019; Chahine et al., 2018; Haeffelin et al., 2005; Zhang et al., 2019). Located at around 15 km southwest of the Paris administrative borders, the SIRTA is considered as a peri–urban site due to its lower population density in an environment mixing forest, urban areas as well as agriculture fields and traffic roads. Therefore, measurements carried out at SIRTA have been classified as
background values for the Paris area (Bedoya-Velásquez et al., 2019). Full instrumentation available at SIRTA for long−term measurements is described at https://SIRTA.ipsl.fr/data−overview/ (last access: 14 June 2024).

The Rambouillet forest ground–based supersite (48.6866° N, 1.7045° E; hereafter RambForest), was hosted at the "La Boissiere–Ecole" French commune, located in the western part of the Rambouillet forest, within the Ile−de−France region and at around 50 km southwest of the Paris administrative borders. The site covers a surface area of 625 m$^2$ and include a 40
m high tower, originally dedicated to the surveillance of forest fires above the canopy (about 25 m high). The area is surrounded by the Rambouillet national forest, a mixed (pine and oak, primarily) deciduous and evergreen trees forest. A large panel of instrumentation was installed below and above the canopy, from the ground to the top of the tower, in order to measure biogenic and anthropogenic VOCs, gas pollutants, the main oxidants and aerosols. Measurements below the canopy relevant to this paper were performed with the PEGASUS mobile facility (PortablE Gas and Aerosol Sampling UnitS,
https://pegasus.aeris−data.fr/, last access: 6 June 2024) (Formenti et al., in preparation) and the mobile facility BARRACUDA (Kammer et al., 2020).

### 2.2 In situ instrumentation

The optical CRI retrieval and SSA calculations are based on measurements of the aerosol light absorption and scattering coefficients ($\beta_{abs}$, $\beta_{sca}$, units of Mm$^{-1}$) and the particle number size distribution (units of particles per cm$^3$), as it will be described
in Section 3. Here, we describe the different instruments providing these observables at PRG, SIRTA and RambForest, the data treatment and evaluation of their uncertainties. Aerosol measurements at the three sites were performed with a submicron (<1 μm) particulate matter (PM$_1$) certified sampling head. Data presented in this paper were averaged over 1−hour and reported at Standard Temperature and Pressure (STP) conditions (T=273.15 K and P=1013.25 hPa).

#### 2.2.1 Spectral aerosol absorption coefficient

Aerosol light absorption coefficient at seven wavelengths (370, 470, 520, 590, 660, 880, and 950 nm), together with equivalent black carbon (eBC) concentration, were measured at 1−min resolution by the dual–spot Aethalometer (AE33) Magee (Drinovec et al., 2015) for all the sites under investigation. The AE33 measures the light attenuation coefficient $\beta_{atn}$ through



a filter–based optical method. The absorption coefficient is calculated as $\beta_{abs}=\beta_{atn}/C_{ref}$, where $C_{ref}$ is the multiple scattering correction parameter, accounting for the multi−scattering by the filter fibers in the instrument (Collaud Coen et al., 2010;

Weingartner et al., 2003). In this paper, we used the spectral invariant $C_{ref}$ ($C_{ref} = FF \bullet H^* = 2.45$, where $H^*=1.76$, an harmonization factor, and $FF =1.39$, the base $C_{ref}$ provided by the manufacturer for the M8060 filter type) recommended by the pan−European ACTRIS research infrastructure (ACTRIS, 2023; Savadkoohi et al., 2023). However, literature shows a large variability of $C_{ref}$, depending on the filter type but also potentially linked to the aerosol composition, mixing state and SSA, with values ranging between 2 and 5 (Fig. S1) (Drinovec et al., 2022; Kalbermatter et al., 2022; Bernardoni et al., 2021;

Moschos et al., 2021; Yus-Díez et al., 2021; Valentini et al., 2020; Drinovec et al., 2015). In order to account for this variability (major source of uncertainty in aethalometer absorption measurements), a wavelength–independent average $C_{ref}$ from the ensemble of literature values depicted in Fig. S1 was estimated as 3.38. This value provides 38% deviation from the ACTRIS reference $C_{ref}$ value of 2.45 (used in this study). In order to account for this deviation, we assume that 38% is the uncertainty on $\beta_{abs}$ due to the multiple scattering correction. An additional uncertainty of 10% due to noise was assumed from Cuesta-

Mosquera et al. (2021) to account for the instrumental noise. The variation due to temporal average, calculated as the standard deviation divided by the mean of the hourly average value, was estimated to contribute from 30% at 370 nm up to 44% at 950 nm at the three sites. The overall uncertainty on $\beta_{abs}$ was calculated as the square root of the sum of the squared uncertainties due to noise, temporal variation and $C_{ref}$, reaching values up to 50% at all the three sites and at all wavelengths.

The spectral variability of the aerosol absorption coefficient was parametrized by the Absorption Angstrom Exponent (AAE)

calculated as described in Eq. (1).

$$AAE = \ln\left(\frac{\beta_{abs}(\lambda_1)}{\beta_{abs}(\lambda_2)}\right)/\ln\left(\frac{\lambda_1}{\lambda_2}\right), \qquad \lambda_1 = 370 \text{ nm}, \lambda_2 = 950 \text{ nm} \qquad\qquad (1)$$

### 2.2.2 Spectral aerosol scattering coefficient

The aerosol spectral light scattering coefficient was measured by three different models of nephelometers (one per sites) measuring at a temporal resolution between 1 and 5 s and operating at different wavelengths: the Ecotech Aurora 4000 at PRG

(450, 525 and 635 nm; (Teri et al., 2022)), the Ecotech Aurora 3000 at SIRTA (450, 525 and 635 nm; (Teri et al., 2022)), and the TSI 3563 at RambForest (450, 550 and 700 nm; (Anderson and Ogren, 1998)). The nephelometer geometry measure the scattering coefficient within a specific angular range, excluding near−forward and near−backward scattering. This introduces an underestimation known as truncation error. Both the Aurora 3000 and 4000 measure $\beta_{sca}$ at the angle range $\theta_1-\theta_2=9°−$ 170°, while the TSI 3563 measures within $\theta_1-\theta_2=7°−170°$. This truncation error was corrected following the formulations for

submicron aerosols provided by Teri et al. (2022) for the Ecotech Aurora 4000, Müller et al. (2011) for the Ecotech Aurora 3000, and Anderson and Ogren (1998) for the TSI 3563, based on the knowledge of the Scattering Angstrom Exponent (SAE), representing the spectral variation of $\beta_{sca}$. Identifying $\beta_{sca,no-tc}$ as the scattering coefficient measured by the instrument without truncation corrections, the SAE was calculated as shown in Eq. (2).

$$SAE = \ln\left(\frac{\beta_{sca,no-tc}(\lambda_1)}{\beta_{sca,no-tc}(\lambda_2)}\right)/\ln\left(\frac{\lambda_1}{\lambda_2}\right), \qquad \lambda_1 = 450 \text{ nm} \quad \lambda_2 = 635 \text{ or } 700 \text{ nm} \qquad (2)$$

The estimated truncation correction varied on average between 10% and 4% in the 450−700 nm nephelometer measurement range. In this paper, we will use $\beta_{sca,no-tc}$ to refer to native nephelometer measurements (non−corrected for truncation), while the truncation–corrected scattering coefficient will be referred as $\beta_{sca}$ and it will be used to investigate the aerosol scattering coefficient variability and estimate the single scattering albedo (see Section 3.2). The aerosol light scattering coefficient was extrapolated at the seven wavelengths of the AE33 using the SAE and applying the Eq. (3).

$$\beta_{sca}(\lambda_x) = \beta_{sca}(\lambda_2) \cdot \left(\frac{\lambda_x}{\lambda_2}\right)^{-(SAE)}, \quad \lambda_x=370, 470, 520, 590, 660, 880, 950 \text{ nm} \qquad (3)$$



The uncertainty on $\beta_{sca}$ was calculated as the quadratic sum of the noise and standard deviation uncertainties (neglecting uncertainties due to spectral extrapolation, drift and truncation correction). Uncertainty due to noise was assumed to be 3% for the Ecotech Aurora 3000 and 4000 (Teri et al., 2022) and 10% for the TSI 3563 (Anderson et al., 1996). The uncertainty due to hourly temporal average, calculated as the standard deviation divided by the mean of the hourly average value, was estimated to contribute from 6% at 370 nm up to 37% at 950 nm at the three sites. The overall uncertainty on the $\beta_{sca}$ was estimated to range within 7% and 39%.

### 2.2.3 Aerosol number size distribution

The aerosol number size distributions were provided by a scanning mobility particle sizer (SMPS, TSI) and an optical particle counter (OPC, GRIMM Inc.,). The SMPS classifies and counts particles which are selected based on their electrical mobility diameters ($D_m$). Particles flowing into the SMPS system are firstly neutralized (here using a soft X−Ray radioactive source), secondly classified by their size according to their electrical mobility through a Differential Mobility Analyzer (DMA model 3081 for PRG and SIRTA, and 3080 for RambForest site) and counted by a Condensation Particle Counter (CPC model 3775 for PRG, 3010 for SIRTA, and 3772 for RambForest). The SMPS can provide counting up to 1 μm of mobility diameter with a time resolution of 3−min. The GRIMM OPC classifies particles based on their optical diameter ($D_{opt}$) defined by the light scattering signal intensity derived from interaction with a monochromatic light source (Heim et al., 2008) and works at a temporal resolution of 6 s. During the ACROSS campaign, three SMPS were deployed (one per sites) with slightly different configurations, leading to diverse sampled $D_m$ ranges (23.3–982.2 nm at PRG, 8.9–829.0 nm at SIRTA, 19.5–881.7 nm at RambForest; note that RambForest data are available only from the 27$^{th}$ of June onward), while the GRIMM OPC was deployed at PRG (GRIMM 1.108; optical diameter range 0.3–20 μm, 780 nm operating wavelength) and RambForest (GRIMM 1.109 model; optical diameter range 0.25−32 μm, 655 nm operating wavelength), but not at SIRTA.

The $D_m$ measured by the SMPS can be converted into geometrical (or volume equivalent) diameter $D_g$, by dividing for the dry dynamic shape factor (DeCarlo et al., 2004), set in our study to 1 in the assumption of spherical particles. The $D_{opt}$ measured by the OPC can be converted into $D_g$ by the knowledge of the CRI of the aerosol at the operating wavelength of the instrument (Formenti et al., 2021; Heim et al., 2008). As it will be described in Section 3.1, the optical to geometrical OPC conversion consists in deriving the real part of the unknown refractive index for sampled aerosols at PRG and RambForest.

The uncertainty of the particle number size distribution (dN(Dp)/dlogDp) was calculated for both each SMPS and the OPC as the counting uncertainty due to Poisson statistics ($\Delta dN(Dp) = \pm\sqrt{dN(Dp)}$), resulting in an average 10% uncertainty in the submicron range. The total number concentration $N_{tot}$ in the PM$_1$ fraction was calculated by summing up number concentrations from SMPS size bins.

### 2.3 Ancillary measurements and products

Together with the relevant optical and size measurements, other ancillary data are considered in this work:

- meteorological observations, including temperature, pressure, wind speed, wind direction, and mixing layer height (MLH), were obtained by in–situ meteorological station and from attenuated backscatter signals measured by a ceilometer at the SIRTA site. Data from the SIRTA observatory, located in the middle between the urban and the forest sites, are considered as representative of average meteorological conditions between the three sites;

- aerosol non–refractory chemical composition (organic, nitrate, sulfate, ammonium, chloride) in the PM$_1$ fraction was measured by a Time–of–Flight Aerosol Chemical Speciation Monitor (ToF–ACSM, Aerodyne Research, 6−min resolution) at PRG, by a quadrupole ACSM (Q−ACSM, 29−min resolution) at SIRTA sites and by a High−Resolution ToF Aerosol Mass Spectrometer (HR−ToF–AMS, Aerodyne Research, 3−min resolution) at the RambForest site.



Each of these three instrument types is produced by Aerodyne Res. Inc. (Billerica, USA). Their measurement
        principles are fully described by Fröhlich et al (2013), Ng et al. (2011) and DeCarlo et al., (2006) respectively. Based
        on previous dedicated studies, uncertainties on the total non−refractory mass concentration from the ACSM and AMS
        is evaluated to be around 20−35% (Budisulistiorini et al., 2014; Crenn et al., 2015; Middlebrook et al., 2012).

   -    the black carbon (BC) Paris–to–regional ratio ($BC_{ratio}$ (Di Antonio et al., 2023a)), is a product derived from the
WRF−CHIMERE v2020r3 CTM (chemistry transport model, Menut et al., 2021) simulation for the ACROSS
        campaign 2022, also described in (Di Antonio et al., 2024a). The CHIMERE model is a 3D regional Eulerian CTM,
        used to simulate gas and aerosols concentrations. The WRF meteorological model (v.3.7.1, forced with the NCEP
        initial and boundary conditions) is coupled with CHIMERE and provides the input meteorological fields. The
        anthropogenic emissions (e.g. BC, OC) are provided by the CAMS−GLOB−ANT product with a spatial resolution
of 0.1°x0.1° (Soulie et al., 2023). Three nested domains were configured respectively at 30 km, 6 km and 2 km spatial
        resolution and centered over the Ile−de−France region. Aerosol dry deposition was simulated according to Zhang et
        al. (2001), and wet deposition below−clouds due to rain according to Willis and Tattelman (1989).

        The $BC_{ratio}$ product (retrieved only for the simulation at 2x2 km spatial resolution) aims at tracing the modelled BC
        emissions in the Grand Paris domain (Grand-Orly Seine Bièvre, 2018) compared to the regional domain (Fig. 1), in
order to be able to identify periods where the SIRTA and RambForest sites were under the influence of the Paris air
        masses. A tracer of BC from Grand Paris area emissions ($BC_{Paris}$) was introduced, such as the total black carbon
        concentration can be split into the Paris and a regional contribution, i.e. $BC_{tot}= BC_{Paris} + BC_{regional}$. The $BC_{ratio}$
        is calculated as follows:

$$BC_{ratio} \,(lon, lat, time) = \frac{BC_{Paris}(lon,lat,time)}{BC_{Paris}(lon,lat,time)+BC_{regional}(lon,lat,time)} \tag{4}$$

where, for each pixel within the considered domain, the $BC_{regional}$ accounts for the BC simulated concentrations
        without taking into account the Grand Paris area BC emissions (i.e. with BC emissions within the Grand Paris area
        equal to zero), while $BC_{Paris}$(lon, lat, time) represents the simulated BC concentrations from the Grand Paris area
        emissions only. A value of 1 of the $BC_{ratio}$ indicates that all the BC is due to simulated concentrations from the Grand
        Paris area. On the contrary, a value of 0 in the $BC_{ratio}$ means that no contribution from Grand Paris area is traced.
Intermediate values weight the contribution of the Grand Paris area over the total BC concentration simulated in the
        regional domain. An example of the $BC_{ratio}$ useful to trace the evolution of the Grand Paris area plume is shown in
        Fig. 1.

   -    The Brown Carbon (BrC) contribution to the 370 nm absorption coefficient ($\beta_{abs-BrC}$) was calculated following Eq.
        (4) in Zhang et al. (2020b) as shown in Eq. (5).
$$\beta_{abs-BrC}(370nm) = \beta_{abs}(370nm) - \beta_{abs-BC}(370nm) \tag{5}$$
        The term $\beta_{abs-BC}(370nm)$ in Eq. (5) is the BC absorption contribution to the total absorption at 370 nm
        parametrized as described in Eq. (6).
        $$\beta_{abs-BC}(370nm) = \beta_{abs}(880nm) \cdot (\tfrac{370nm}{880nm})^{-AAE_{BC}} \tag{6}$$
        where $AAE_{BC}$ is the black carbon AAE, and assumed to be 1.
It is important to note that the absorption coefficient that we associate to brown carbon could be affected by the fine
        dust aerosol absorption too. However, as the mass absorption coefficient (MAC) of dust is significantly lower
        compared to that of BrC (Samset et al., 2018; Caponi et al., 2017), and as we did not expect substantial sources of
        fine dust in proximity of the ACROSS sites, we attributed the absorption coefficient retrieved following Eq. (5)
        predominantly to BrC.



## 3. Spectral aerosol CRI and SSA assessment

### 3.1 CRI retrieval procedure

Hourly CRI at the seven wavelengths of the AE33 (370, 470, 520, 590, 660, 880, 950 nm) were retrieved using the iterative method illustrated in Fig. S2 and detailed in Text S1. This method consists in identifying n and k values that allows, by optical calculation using the measured number size distribution as input, to reproduce at each wavelength the measured spectral $\beta_{sca}$ and $\beta_{abs}$. Calculations were made under the assumption of homogeneous spherical particles (Mie theory) and considering nephelometer measurements, non−corrected for truncation, given that the truncation correction depends on the a priori assumption on the aerosol complex refractive index.

Size distribution measurements from the SMPS were used as input to the calculations. The n and k free parameters were varied in the [1.2, 2.0] range for n (at 0.01 step) and [0, 0.02] for k (at 0.001 step) and the optimal (n, k) pairs were determined by minimizing at each wavelength the root mean squared difference (RMSD) between observed and modeled $\beta_{sca}$ and $\beta_{abs}$ calculated as shown in Eq. (7).

$$RMSD = \sqrt{\left(\frac{\beta_{sca-obs} - \beta_{sca-mod}}{\beta_{sca-mod}}\right)^2 + \left(\frac{\beta_{abs-obs} - \beta_{abs-mod}}{\beta_{abs-mod}}\right)^2} \tag{7}$$

Sensitivity calculations were performed to evaluate the uncertainties on the retrieved n and k values in relation to the uncertainty on the input parameters such as: i) the instrumental error due to noise and the standard deviation of the input and ii) the dataset variability (median and percentiles of the inputs) as detailed in Table S1. At each hourly time step and each wavelength, the results are reported as average CRI ± SD (standard deviation) of the retrieved values from the different sensitivity simulations.

In addition to the iterative method described above, for PRG and RambForest where OPC measurements were also available, a second approach, named OPC−SMPS overlap method, was used to derive an additional estimate of the real part of the CRI. This approach, detailed in Text S2 and Text S3, is based on the fact that the conversion of the OPC aerosol number size distribution from optical diameters $dN(D_{opt})/dlogD_{opt}$ to geometrical diameters $dN(D_g)/dlogD_g$ requires the knowledge of the complex refractive index. Comparing the OPC and SMPS measurements in their overlap region, and minimizing their differences by varying the optical−to−geometric conversion (which depends on the CRI), allow retrieving information on the CRI at 780 nm (PRG) and 655 nm (RambForest), 780 nm and 655 nm being the operating wavelengths of the OPC installed at the two sites, respectively. In order to apply this method, look up tables for OPC optical−to−geometrical diameter conversion values reported by Formenti et al. (2021) were considered, providing data in the n range [1.33,1.75] (at 0.01 step) and in the k range [0,0.4] (at 0.001 step), both at 655 nm and 780 nm. As the OPC optical−to−geometric corrections below the diameter of 1 µm are mostly sensitive to n and less sensitive to k variations (see Fig. S3 and Fig. S4), this method does not constrain both the real and imaginary components but only n, while k is kept fixed at the spectral value of the optical−closure retrieval. The OPC−SMPS overlap method, already applied in previous studies (Flores et al., 2009; Hand and Kreidenweis, 2002; Mack et al., 2010; Vratolis et al., 2018), was employed here specifically with two objectives. First, by comparison with the iterative method, it was employed to establish a minimum aerosol load threshold above which the refractive index retrieval can be considered satisfactory (as discussed in Text S3). In this way, all points below the threshold were effectively discarded as the aerosol load is considered not sufficiently high to perform the retrieval. Results of the optical−iterative vs OPC−SMPS method comparison (Fig. 2) suggest that a mass threshold of 3 µg.m$^{−3}$ for CRI retrieval, corresponding to a $\beta_{ext}=\beta_{sca} + \beta_{abs}$ of about 12 Mm$^{−1}$ (assuming a MEC of 4 m$^2$.g$^{−1}$), is required to obtain well−constrained results. For the present work, the complex refractive index results from the iterative method are reported and discussed assuming this threshold applied. As second objective, the OPC−SMPS overlap method was used to provide a comparison dataset to the iterative method, and therefore to qualify the broad consistency of the retrieved results. By looking at the comparison of n from the two retrieval methods in





Figs. 2c and 2d, a clear correlation is identified, indicating that the retrieved n magnitude and temporal variation is similar between the two approaches. Nonetheless, a systematic difference is evidenced, with the n from the OPC−SMPS retrievals systematically higher than the one from the iterative method, averaging at 25% difference for PRG and 7% difference for RambForest. As it is difficult to assess the causes of these differences, this comparison is taken as an evaluation of the inherent uncertainty in the CRI retrieval based on the number concentration and optical observations in the present work.

### 3.2 SSA calculation

The hourly averaged spectral aerosol SSA was calculated at the seven operating wavelengths of the aethalometer as:

$$SSA\ (\lambda) = \frac{\beta_{sca}(\lambda)}{\beta_{ext}\ (\lambda)} = \frac{\beta_{sca}(\lambda)}{\beta_{sca}(\lambda) + \beta_{abs}\ (\lambda)} \tag{8}$$

where $\beta_{ext}$ is the calculated extinction coefficient as the sum of $\beta_{sca}$ (truncation−corrected) and $\beta_{abs}$. The uncertainty on SSA was calculated with the statistical error propagation formula, as shown in Eq. (9).

$$\Delta SSA = \sqrt{\left(\frac{\beta_{abs}}{\beta_{sca}(\lambda) + \beta_{abs})^2}\right)\Delta\beta_{sca})^2 + \left(\frac{\beta_{sca}(\lambda)}{(\beta_{sca}(\lambda) + \beta_{abs})^2}\right)\Delta\beta_{abs})^2} \tag{9}$$

The $\Delta SSA$ ranges on average within 14–43%, 11%–45%, 17–25% at the PRG, SIRTA and RambForest sites, respectively, between 370 nm and 950 nm.

### 4. Results

### 4.1 Aerosol optical properties during the ACROSS campaign 2022

Figure 3 depicts the time series of $\beta_{sca}$, $\beta_{abs}$, and $\beta_{ext}$ at 520 nm and the total particle number concentration $N_{tot}$, for the three sites in the June 15–July 25 2022 period together with temperature, mixing layer height and wind vector at the SIRTA site. Figure 4 shows the time series of the retrieved n, k, and SSA at 520 nm for the three sites.

The ACROSS campaign was characterized by different atmospheric conditions, mirrored in the optical coefficients and CRI signal variability: i) two strong heatwaves occurred in the June 15–18 and July 12–25 periods, with hourly temperatures above 340 35 °C registered at the peri−urban site during the first heatwave and close to 40 °C during the second heatwave; ii) a "cleaner" period from June 23 to July 11, 2022, characterized by low aerosol loadings and temperatures close to climatological averages; iii) a strong long−range transport fire episode on July 19 2022. Two Saharan dust intrusion episodes (SDI) also occurred during the first and the second heatwaves. Note that for the RambForest site, because of the limited availability of the particle size distribution data and the mass−threshold applied discussed in Section 3, the CRI retrievals are only partially available during 345 the clean and the second heatwave periods. Average aerosol optical properties at 520 nm for the different periods (full period, clean period, heatwaves) are summarized in Table 1. The following paragraphs provide descriptive statistics for each period.

#### 4.1.1 Clean Period (23 June−11 July 2022)

The clean period was marked by low scattering and absorption coefficients at 520 nm (below 40 Mm$^{-1}$). The mixing layer height extended up to 2.5 km, and the total number of particles reached the lowest values of (1.6, 1.2, 0.2) $\cdot 10^3$ cm$^{-3}$ at PRG, 350 SIRTA and RambForest, respectively, as averages during the period. The imaginary CRI component at 520 nm varied from 0.01 to 0.16 between the three sites, reaching the maximum value at the SIRTA site, and the minimum at RambForest. The SSA minimum value of 0.38 at 520 nm was registered at PRG (urban site). On the contrary, a maximum of 0.96 for the SSA was measured at RambForest (forest site). An average SSA absolute difference of around 0.2 was estimated between the two sites during the clean period.



### 4.1.2 First (15–18 June 2022) and second (12–25 July 2022) heatwaves

The heatwave periods reflected in the highest scattering and absorption coefficients and particle number concentrations during the campaign. Temperatures above 35°C and highly developed mixing layer heights (maximum value of 3 km at the SIRTA site) were observed. The extinction coefficient at 520 nm maximized at values between 40 and 80 Mm$^{-1}$ during the first heatwave at the three sites, reaching up to 150 Mm$^{-1}$ during the second one (fire event excluded), and N$_{tot}$ reached maximum values of (22, 19, 9) •10$^3$ particles cm$^{-3}$ at PRG, SIRTA and RambForest sites respectively. The SSA reached values up to 0.95 at the urban site, and the average absolute differences with the SSA at the RambForest site reduced to around 0.1 compared to the clean period. The minimum SSA values (~0.43) were registered during the night of 13 July in correspondence of the fireworks occurring in the suburban Paris area in occasion of the French National Day celebrations. The imaginary CRI component was on average lower at the forest site (lowest k value of 0.06 at 520 nm), compared to the urban and peri−urban sites (highest k value of 0.13 at 520 nm). The k was higher during the second heatwave period compared to the first one, as observed at PRG and SIRTA (no CRI retrieval available for RambForest during the first heatwave).

During the heatwaves, two main Saharan dust transport events were also reported (observed from Lidar measurements at SIRTA and with an increase of the PM$_{10}$ surface observations, not shown), causing dust intrusions from the free troposphere to the surface layer. These episodes were marked by a decrease in the SAE signal showing a minimum value of 1.9 on June 18 (first heatwave) compared to the 2.3 average value at the urban site, but not clearly distinguishable in absorption and scattering coefficient absolute signals, likely due to the PM$_1$ sampling head cut−off not efficient in sampling coarse dust.

### 4.1.3 The fire episode of 19 July 2022

An exceptionally intense event of long−range transport of biomass burning aerosols (identified as fire episode, FE) from southern France (in the Landes forest, (Menut et al., 2023)) to the Ile−de−France region occurred in the evening of July 19 (the fire plume arrived at around 17:00 UTC at the RambForest site, and at around 18:00 UTC at the urban site), corresponding to the hourly maxima of (340, 203, 253 Mm$^{-1}$) for $\beta_{sca}$ and (33, 22, 25 Mm$^{-1}$) for $\beta_{abs}$ at 520 nm for the PRG, SIRTA and RambForest sites, respectively (See Fig. 3). This episode was documented in the recent CAMS2_71 report N°04 in 2022 (Tsyro et al., 2023).

Average hourly CRI values of 1.53−0.018i (PRG) and 1.57−0.020i (RambForest) at 520 nm were retrieved for the fire episode, showing an increase in the real part component and a decrease in the imaginary part, as compared to the average values of the period (size distribution data not available at the peri−urban SIRTA site for CRI retrieval).

The AAE during the fire episode of July 19 reached maximum values of 1.64, 1.60 and 1.75 at the urban, peri−urban and forest sites, respectively, significantly larger than the AAE<1.29 values reported for the heatwave periods, suggesting the contribution of brown carbon absorption at shorter wavelengths. The fraction of absorption due to BrC ($\beta_{abs-BrC}$) at 370 nm calculated following Eq. (5) is also reported in Table 1. The $\beta_{abs-BrC}$ significantly increased during the fire episode reaching hourly maximum values of 31, 19 and 29 Mm$^{-1}$ and representing 45%, 42% and 52% of the total absorption coefficient at 370 nm during the fire episode, significantly higher than the estimated full period averages of 11%, 17% and 22% for the urban, peri–urban and forest sites respectively. As previously discussed, fine dust aerosols could contribute to the $\beta_{abs-BrC}$ absorption signal. We estimate that during the first heatwave period, where we expected the possible stronger contribution of fine dust aerosols in the submicron fraction compared to biomass burning aerosols (due to documented Saharan dust intrusion at the ground from the free troposphere, not shown), the $\beta_{abs-BrC}$ reached maximum values of 4.6, 4.4 and 5.2 Mm$^{-1}$ at the three sites, respectively, significantly lower than the values observed during the fire event.



**4.2 CRI and SSA diurnal cycle**

CRI and SSA show different diurnal cycles at the different sites, as illustrated in Fig. 5. At both urban and peri–urban locations
n showed maximum values in the early afternoon at around 13:00–15:00 UTC, and lower values during the night. Together with higher values during the early afternoon, the real refractive index at the forest site also showed an additional local maximum during the night. Concerning the CRI imaginary part, both urban and peri–urban sites showed morning and late afternoon peaks, suggesting the role of BC emissions due to traffic activity in affecting the imaginary CRI component. Moreover, a local maximum value of k is identified at the forest site, but few hours later than at the urban and peri–urban sites.
The SSA showed the opposite behavior compared to k, with the highest values obtained during the early afternoon (corresponding to maximal photooxidation activity), while lower values reached over the peak in absorption due to traffic emissions in the early morning and in the late afternoon.

**4.3 CRI and SSA spectral variability**

Both scattering and absorption coefficients decreased with λ at the PRG, SIRTA and RambForest sites (Fig. S5). Slight
variations were observed during the heatwaves (the AAE increased only at the urban and forest sites; the SAE increased only at the peri−urban site) and the clean period (Table 1, Fig. S6). The AAE values and gradients are in line with the previous observations reported by Favez et al. (2009) who retrieved an average AAE value of 1.07 for summertime in Paris (based on air quality observations in the same district as the PRG station).

Figure 6 shows the CRI and SSA spectral variabilities at the three sites for the whole ACROSS campaign. Full period averages
showed a slight wavelength−dependent CRI real component, with decreasing values in the 370−660 nm range (the highest spectral variability being at RambForest), in opposition to a mostly flat or slightly increasing CRI imaginary component with λ. The real CRI temporal full period averages ranged within 1.45−1.41, 1.58−1.54, and 1.62−1.48 between 370 and 950 nm for the urban, peri−urban and forest sites, respectively. For the imaginary component, the highest spectral values of 0.040 were retrieved at the urban and peri−urban sites at 950 nm, whereas a lower value of 0.028 was estimated at the forest site for the
same wavelength. The SSA decreased sharply within the wavelength at all sites, going from values of 0.81, 0.83, 0.90 at 370 nm to 0.58, 0.74, 0.69 at 950 nm for the urban, peri−urban and forest sites, respectively. The spectral behaviour identified for the full average period for both n, k and SSA, remained unchanged during the heatwaves and clean periods (Fig. S8).

As shown in Fig. 6, during the July 19 fire−plume, the k increased at 370 and 470 nm, due to the possible increase in BrC contribution, in particular at the urban site where the derived k ranges between 0.022 and 0.019 in the 370−950 range. On the
contrary, no significant changes in k were observed at the forest site. Since, the FE plume travelled more than 400 km to North, it can be expected to be photochemically−aged, reducing its absorption due to bleaching effects (Laskin et al., 2015). As a matter of fact, some previous studies show that the BrC refractive index may vary between 0.06 and 0.5 at 370 nm for biomass burning aerosols (Runa et al., 2022; Sumlin et al., 2018; Liu et al., 2015; Lack et al., 2013; Chakrabarty et al., 2010; Alexander et al., 2008; Kirchstetter et al., 2004).

**5. Discussion**

**5.1 CRI and SSA link to meteorological conditions**

In this section, we analyse the submicron (PM$_1$) single scattering albedo and complex refractive index variability at the three sites, under different meteorological conditions. As illustrated in Fig. 3 and Fig. 4, the entire June−July 2022 period was characterized by a strong temperature increase above 35 °C, favouring only for few cases the MLH development up to 3 km,
and weak winds (average values of 2.65 m s$^{-1}$ and 2.46 m s$^{-1}$ during the first and second heatwaves, respectively), preventing the pollutant dispersion. A diagram presenting the SSA and CRI averages at 520 nm over the different periods (i.e. clean and heatwaves) is available in Fig. 7, while an analysis by wind sector is summarised in Table 2. Figure 7 shows that on average,





an urban−to−rural gradient is present in SSA and CRI under the different campaign periods. Furthermore, as depicted in Table 2 (and detailed in Table S4), two main regimes can be identified. Under N, NE, E, SE wind conditions, a positive urban−to−rural gradient in SSA can be identified ($\Delta$SSA ($SSA_{RambForest}-SSA_{PRG}$) ~0.1), corresponding to the largest k at SIRTA, intermediate values in PRG and lowest values in RambForest. Under S, SW, W and NW conditions, the SSA positive gradient increases to ~0.2 and a systematic negative urban−to−rural gradient of k is identified. The NE wind directions represent the best wind scenarios to investigate the mixing between the Paris urban plume and the biogenic emissions of the forest site, as the latter one is located NW of Paris. These conditions are driven by a strong high−pressure system over the Great Britain, characteristic of part of the two heatwave periods, leading to a potential advection of the urban air masses to the forest site, including both the fresh Paris emissions and the possible aged aerosols from upwind areas (e.g. the Benelux area). Previous observations during the MEGAPOLI campaign suggested that the regional aerosol advection dominates over the locally generated $PM_{2.5}$ (summertime) fraction in the Paris area (Beekmann et al., 2015; Bressi et al., 2014; Freutel et al., 2013). Indeed, the average CRI imaginary component increases at the peri−urban and forest sites (explained by the absorbing Paris pollution plume), while it decreases at the urban, under NE conditions compared to S/SW/W/NW directions, due to advection of aged continental background aerosol.

Under northerly winds, more polluted air masses may be advected over the Paris area, originating from the Great Britain (Chazette et al., 2005). Average SSA and CRI values of (0.73, 0.79, 0.86) and (1.40−0.035i, 1.56−0.039i, 1.48−0.021i) were observed during the campaign under these conditions, for the urban, peri−urban and forest sites, respectively. Under SE and E wind directions, the strongest (lowest) average k (SSA) values were observed at the forest (0.053 and 0.73) and peri−urban sites (0.032 and 0.80), respectively, while no significant increase was observed at the urban site. Conversely, the forest and peri−urban sites showed the lower average CRI of 1.53−0.024i and 1.51−0.021i, respectively, under south−westerly winds (air masses coming in this case from the Loire Valley and SW France regions). Considering only the westerly, south−westerly, and southerly winds (representing the urban as downwind and the forested site upwind), cleaner air masses are expected (e.g. originating from the Atlantic Ocean under westerly winds). This is the case for the peri−urban and forest sites where higher wind sector average (W/SW/S) SSA of (0.84, 0.89) and low CRI of (0.026, 0.019) were observed compared to other wind regimes, suggesting a cleaner environment, more dominated by scattering aerosols (e.g. marine or biogenic aerosols). On the contrary, the SSA (CRI) reaches the lowest (highest) values of 0.66 (0.051) at the urban site in particular during the first part of the clean period, suggesting a stronger influence of local anthropogenic emissions (BC−dominated) under low aerosol loading.

Finally, the strongest heatwave days (17−18 June and 19 July 2022), are mainly characterised by southerly winds, and are associated to the maximum MLH value during the campaign of nearly 3 km. The SSA maximum values at 370 nm of 0.93, 0.95 were observed respectively at the peri−urban and forest sites, reflecting the expected biogenic SOA enhancement with high temperatures in the SSA measurements. As a point of comparison, Dingle et al. (2019) calculated SSA values of 0.98−0.99 at 375 nm for purely biogenic compounds, while aromatics−derived SOA show lower SSA in the 0.75−0.95 range, more typical of the urban environment where a maximum value of SSA of 0.90 at 370 nm was measured during the heatwave period.

The real and imaginary CRI difference among the sites as a function of the $BC_{ratio}$ and coloured by the main wind sectors is shown in Fig. 8. Following the previous discussion, for N to SE wind sectors the $|\Delta n|$ and $|\Delta k|$ at 520 nm tends to be zero for the SIRTA and RambForest sites, supporting the hypothesis that the Paris area and the regional background influence the peri−urban and forest sites under these wind conditions. The $|\Delta n|$ and $|\Delta k|$ depicted in Fig. 8, shows that, under N to SE wind sectors, the advected regional pollution and the Paris emissions may homogeneously affect the CRI over the Ile−de−France region. On the other hand, for S to NW directions, the highest (lowest) $|\Delta n|$ and $|\Delta k|$ ($BC_{ratio}$) were observed, suggesting a non−uniform aerosol spatial distribution across all three sites under these weather conditions. Indeed, combining the temporal variability of the imaginary component with the BC Paris−to−regional ratio (as shown within the animation illustrated in the



Supplementary Material), it is possible to observe how the difference between the imaginary components among the sites tends
to be reduced under the Paris influence. On the opposite conditions, the highest differences in refractive index are observed.

**5.2 Comparison of retrieved CRI and SSA with the literature**

Figure 9 shows the comparison of the spectral CRI retrievals for ACROSS (retrieved for the submicron aerosol distribution)
with ambient CRI reported in the literature, including *in situ* and airborne observations and both works conducted in urban and
rural environments, as well as the previous works in the Paris area. Additionally, AERONET columnar retrieval in the
Ile−de−France region for the ACROSS periods are shown. In general, the real part of the CRI retrieved in our study at the
three sites is in agreement with the literature observations, while the range of k values sits in between literature observations
and AERONET retrievals in the Ile−de−France region during the ACROSS campaign. The lower AERONET k values
compared to the existing literature can be explained by the fact that AERONET is a columnar−integrated retrieval and therefore
also representative of a wider area compared to *in situ* observations (measurements being performed at different elevation
angles) (Chen et al., 2020; Di Antonio et al., 2023g; Schutgens, 2020).

The *in situ* CRI retrievals over the Paris area were already performed during the ESQUIF (Etude et Simulation de la QUalité
de l'air en Ile−de−France) and LISAIR (Lidar pour la Surveillance de l'AIR) campaigns, respectively in July 2000 and May
2005 by the synergy of lidar, sunphotometers and *in situ* measurements from Raut and Chazette (2007, 2008) also reported in
Fig. 9. A more detailed version of the figure is reported in the Supplementary material, Fig. S28. Values of 1.56–0.034i at 355
nm and 1.59–0.040i at 532 nm were obtained over the Paris town hall for May 2005 (Raut and Chazette, 2007), while averages
of 1.51−0.017i were obtained from Raut and Chazette (2008) from aircraft measurements over the Paris area. The real part of
the CRI in Raut and Chazette (2007, 2008) is higher than the values retrieved at the urban PRG site, while showing comparable
values with the peri−urban background and forest sites. The imaginary component in the present analysis reflects the range of
variability observed from Raut and Chazette (2007) and from the studies shown in Fig. 9 from airborne and  in situ observations
(Aldhaif et al., 2018; Ebert et al., 2002, 2004; Espinosa et al., 2019; Ferrare et al., 2006, 1998; Hand and Kreidenweis, 2002;
Müller et al., 2002; Redemann et al., 2000; Shingler et al., 2016; Yamasoe et al., 1998; Zhang et al., 2016, 2013).

The SSA obtained at the urban site shows lower values compared to the 0.82 and 0.93 values at 532 nm found over the Paris
area respectively on the 18 May 2005 and during the ESQUIF aircraft campaign (Raut and Chazette, 2007, 2008). Also, the
SSA values observed in this study (under westerly winds, W sector in Table 2), range between 0.73 and 0.87 (520 nm) from
the urban to the forest sites, and are therefore lower compared to the 0.92 average values (532 nm) observed for the 31 July
2000 from Raut and Chazette (2007) during ESQUIF aircraft campaign under comparable weather scenarios.

Finally, the comparison reported in this section shows that the retrieved *in situ* CRI and SSA are in the range of variability of
the existing literature values from *in situ* observations. Higher differences in the imaginary part of the refractive index are
observed when comparing to columnar-integrated values. This discrepancy may be attributed to the vertical dilution of aerosol
concentration during the diurnal evolution of the planetary boundary layer height, as well as the impact of the vertical
atmospheric stratification.

**5.3 CRI and SSA vs aerosol bulk chemical composition**

Figure 10 illustrates the aerosol submicron bulk chemical composition measured and classified by several classes on CRI, SSA
and AAE. The major $PM_1$ contributor is represented by the organic fraction, showing average values of (65%, 59% and 68%),
followed by sulfate (17%, 23%, 19%) at the urban, peri−urban and forest sites, respectively. The average eBC relative
contributions represent about 5%, 3% and 2% from PRG to RambForest, respectively. When looking at the variation of the
optical properties in relation to composition, a clear impact of the eBC fractions is apparent on the imaginary CRI and SSA
components, both at the urban and peri−urban sites, with an increase of the eBC fraction up to (14% and 11%) respectively in



the k>0.1 class. Therefore, a positive correlation between the aerosol absorption and the primary anthropogenic emissions is suggested at the urban and peri−urban locations. On the other hand, the highest (lowest) SSA (CRI) values >0.9 (<0.05), is associated to higher concentrations of sulfate and nitrate and a decrease in the organic and eBC fractions, in particular at the urban site. Furthermore, the increase of the AAE at the urban and peri−urban sites is associated to an increase of the organic fraction, suggesting an enhanced absorption of brown carbon at shorter wavelengths is linked with the decrease of inorganic

non−absorbing compounds. Looking at Fig. 10, no clear correlation between the composition and the CRI, SSA, and AAE is observed at the forested site, suggesting a combination of different effects, such as absorption enhancement due to lensing effect with aerosol aging (Zhang et al., 2018) or the contribution of different absorbing secondary BrC species under diverse conditions. Finally, no significant variations in the real refractive index can be directly attributable to the bulk composition.

**6. Concluding remarks**

In this study, we have investigated the aerosol complex refractive index and single scattering albedo datasets retrieved during the ACROSS campaign in June−July 2022, under several contrasting atmospheric conditions, at three different sites representatives of urban (PRG), peri−urban (SIRTA) and forest (RambForest) conditions in the greater Paris area. Data refer to the submicron (PM$_1$) aerosol component, surface−level observations and cover the spectral range 370−950 nm. The CRI and SSA were retrieved from a synergy between *in situ* aerosol optical (scattering and absorption coefficients) and particle

size distribution measurements and optical Mie calculations.

Our results show a clear urban−to−rural gradient in SSA and CRI over different periods, with varying intensity depending on meteorological conditions. The SSA (imaginary CRI) increases (decreases) going from urban to rural site, showing on average a 0.1 (0.01) change between PRG and RambForest at 520 nm (on a full period average). The gradient is reduced under the influence of the Paris emissions on the surrounding sites under north−easterly wind regimes, as supported by the integration of observations and regional modeling products. Mixing of air masses from biogenic and anthropogenic origin are expected

under these conditions, with consequent possible formation of mixed ASOA and BSOA products. On the contrary, a more marked positive urban−to−rural gradient in SSA (0.2 change between PRG and RambForest) is observed when the Paris urban site is downwind of the peri−urban and forest sites.

The advection of an intense fire plume from the south of France caused a strong air quality and visibility degradation over the Ile−de−France region. The SSA at 370 nm (520 nm) increased up to 0.93 (0.91) and the CRI spectral pattern changed at the urban site, showing an increase in the UV−visible wavelengths, characteristic of possible aged−BrC in the plume. In fact, since the fire−plume pictures atmospheric transport over a long distance, photochemically−aging processes may have occurred, reducing smoke aerosol absorption due to bleaching effects (Konovalov et al., 2021; Laskin et al., 2015). The retrieved k is

within 0.022 and 0.019 across 370−950 nm at the urban site during the fire event, while no significant changes in k were observed at the forest and peri−urban sites in correspondence of the smoke plume advection.

The chemical composition analysis shows that the imaginary CRI is related to eBC fractions, suggesting the key role of primary emissions and low eBC concentrations in affecting absorption, as particularly evident at the urban and peri−urban sites.

Nevertheless, as organics represent more than 50% of the aerosol mass at the three sites, an important contribution of brown carbon to spectral absorption is expected. BrC is estimated to contribute on average up to 10% (urban), 17% (peri−urban) and 22% (forest) to the absorption coefficient at 370 nm. A more detailed and advanced analysis is necessary to provide insight organic composition of aerosols at the molecular scale, in order to relate spectral absorption to the presence of different chromophores, as for example nitro−aromatics, at the different sites and under different conditions. Moreover, a more detailed



characterization of the particle mixing state by particle−level *in situ* measurements would be advantageous to better understand the particle chemical and optical evolution at the forest site, where more aged and internally mixed aerosols are expected.

According to the recent Meteo France report (Météo France, 2022), summer 2022 has been identified as "the summer of extremes" due to strong positive temperature anomalies registered during all the summer 2022 period, a strong deficit in precipitation, and the long duration of the heatwaves episodes. The increase in the heatwave frequencies, leads to increased

accumulation of anthropogenic and biogenic VOC emissions, with possible SOA build−up that could have impacted the aerosol spectral optical properties (Cholakian et al., 2019; Gomez et al., 2023; Yli−Juuti et al., 2021). Average conditions at 520 nm show that urban SSA is 12 % higher during the heatwaves compared to the clean period at the urban site, while no significant changes are observed at SIRTA and RambForest. Nevertheless, on average, the CRI imaginary component is higher (respectively by 9% at PRG and SIRTA, and 13 % at RambForest) during the heatwaves compared to the clean period.

Therefore, based on the unique dataset presented in this study, we suggest an average CRI at 520 nm of 1.45−0.043i, 1.56−0.041i, 1.52−0.026i respectively for locations showing urban background, peri−urban and forest features under heatwave conditions, while an average CRI of 1.39−0.039i, 1.52−0.038i, 1.47−0.023i is more appropriate under low aerosol loading scenarios.

**Code availability.**

The CRI optical retrieval code can be made available upon request to the first author. The "miepython" routine was used for Mie theory calculations.

**Data availability.**

Level 2 datasets used in the present study from the ACROSS field campaign for the PRG and RambForest sites are available or will be made soon available on the AERIS datacenter (https://across.aeris−data.fr/catalogue/) and DOI referenced. Some

datasets already available are: the equivalent black carbon at the PRG site (Di Antonio et al., 2023c); the spectral scattering coefficient at the PRG site (Di Antonio et al., 2023e); the spectral absorption coefficient at the PRG site (Di Antonio et al., 2023b); the black carbon ratio (Di Antonio et al., 2023a); the SMPS particle size distribution at the PRG site (Kammer et al., 2024); the OPC particle size distribution at the PRG site (Di Antonio et al., 2023f); meteorological data at PRG site (Di Antonio et al., 2023d): the equivalent black carbon at the RambForest site (Di Antonio et al., 2024c); the spectral scattering coefficient

at the RambForest site (Di Antonio et al., 2024d); the spectral absorption coefficient at the RambForest site (Di Antonio et al., 2024b); the SMPS particle size distribution at the RambForest site (Villenave et al., 2023); the OPC particle size distribution at the RambForest site (Di Antonio et al., 2024e); the non−refractory aerosol composition below and above the canopy at the RambForest site (Ferreira de Brito et al., 2023a, b); the mixing layer height at SIRTA (Kotthaus et al., 2023).
The ACSM non−refractory aerosol composition at PRG will be made soon available on the AERIS website.

The SIRTA observatory data can be downloaded at https://sirta.ipsl.fr/ (last access: 19 July 2024).
The complex refractive index and single scattering albedo datasets will be made available after the paper publication.

**Author contribution.**

LDA and CDB designed the study and discussed the results, in collaboration with MB, JFD, PF, GS and GF. LDA performed the full data analysis. CDB and MB supervised all the analysis work. VM and CC coordinated the ACROSS field campaign.

LDA, CDB, PF, AG, VM, CC, ABa, ABe, MCa, SC, MCi, BDA, JDB, DDH, JRD, SD, OF, PMF, CG, LH, JK, BL, FM, GM, EP, JEP, PA, LP, PP, EDP, VR, KR, MS, EV, and PZ, contributed to the campaign setup, deployment, calibration and operation of the instrumentation, and the data collection and analysis from the PRG, SIRTA and RambForest sites. LDA and CDB led



the writing of the manuscript, with contributions by MB, JFD, PF, GS and GF. All authors commented and reviewed the manuscript.

**Competing interest.**

At least one of the (co−)authors is a member of the editorial board of Atmospheric Chemistry and Physics. Furthermore, one of the co–authors (CC) is guest editor of the Special Issue "Atmospheric Chemistry of the Suburban Forest – multiplatform observational campaign of the chemistry and physics of mixed urban and biogenic emissions". The authors have no other competing interests to declare.

**Special issue statement.**

This article is part of the special issue "Atmospheric Chemistry of the Suburban Forest – multiplatform observational campaign of the chemistry and physics of mixed urban and biogenic emissions". It is not associated with a conference.

**Acknowledgements.**

Useful discussions with M. Mallet, Y. Derimian and J.−C. Raut are gratefully acknowledged. We thank B. Picquet−Varrault,
A. Feron, S. Riley, C. Seto, G. Bergametti, and J.L. Rajot for their contribution to the ACROSS field campaign deployment.

**Funding.**

This work has been supported by the ACROSS and the RI−URBANS projects. The ACROSS project has received funding from the French National Research Agency (ANR) under the investment program integrated into France 2030, with the reference ANR−17−MPGA−0002, and it was supported by the French National program LEFE (Les Enveloppes Fluides et
l'Environnement) of the CNRS−INSU (Centre National de la Recherche Scientifique/Institut National des Sciences de l'Univers). The RI−URBANS project has received funding from the European Union's Horizon 2020 research and innovation program under grant agreement no. 101036245. PEGASUS and SIRTA are national facilities of the CNRS−INSU as part of the French ACTRIS research infrastructure. IMT Nord Europe group has been supported by Labex CaPPA (ANR−11−LABX−0005−01). L. N. Hawkins, E. D. Pronovost, and D. O. De Haan were supported by NSF−IRES 1825094.
This project was provided with computer and storage resources by GENCI at TGCC thanks to the grant 2022–A0130107232 on the supercomputer Joliot Curie's SKL partition.

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





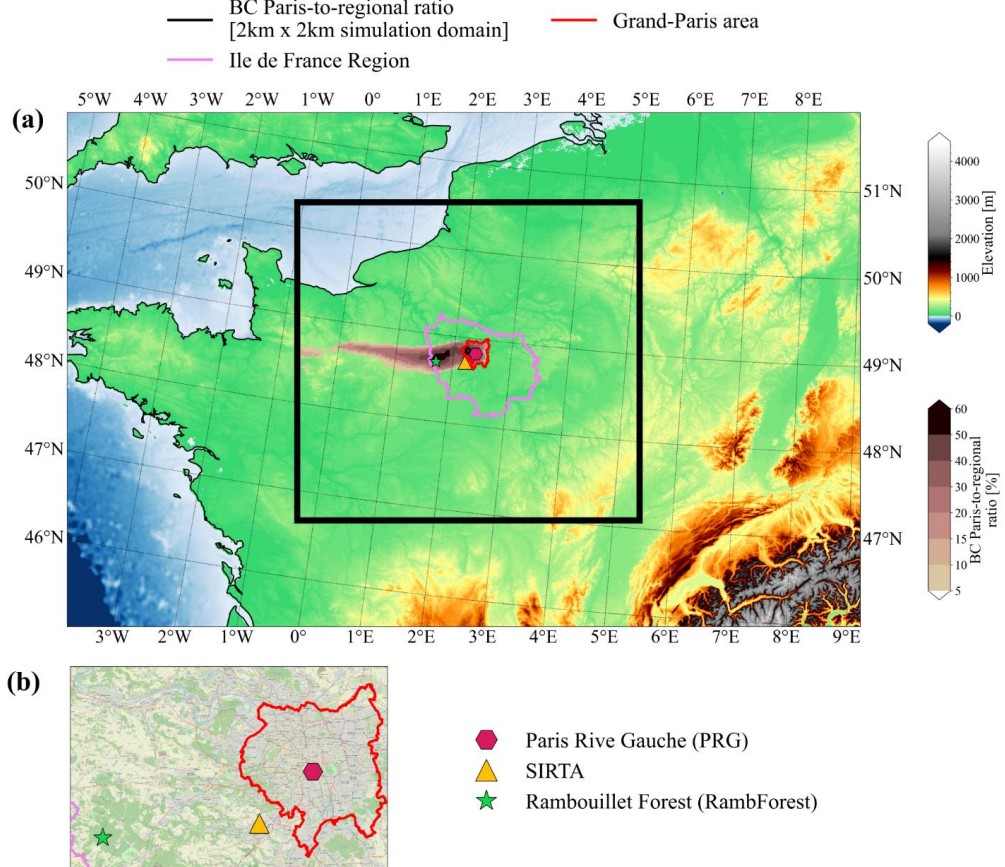


**Figure 1: Geographical location of the Paris Rive Gauche (PRG, urban), SIRTA (peri−urban), and Rambouillet (RambForest, forest) ground−based sites deployed during the ACROSS campaign 2022 in the Ile−de−France region. Panel (a) shows in background the terrain elevation and the BC Paris−to regional ratio (BCratio) for the 22 June 2022 at 13 UTC as an example. The violet line delimits the boundaries of the Ile−de−France region, while the red line delimits the boundaries of the Grand Paris domain. Digital**
**elevation model source: SRTM15 (Tozer et al., 2019a, b). Panel (b) shows a zoom (1.6°−2.62°E, 48.60−49.05°N) over the Ile−de−France region to better visualize the ground−based sites locations. Background map source: © OpenStreetMap contributors 2024. Distributed under the Open Data Commons Open Database License (ODbL) v1.0.**





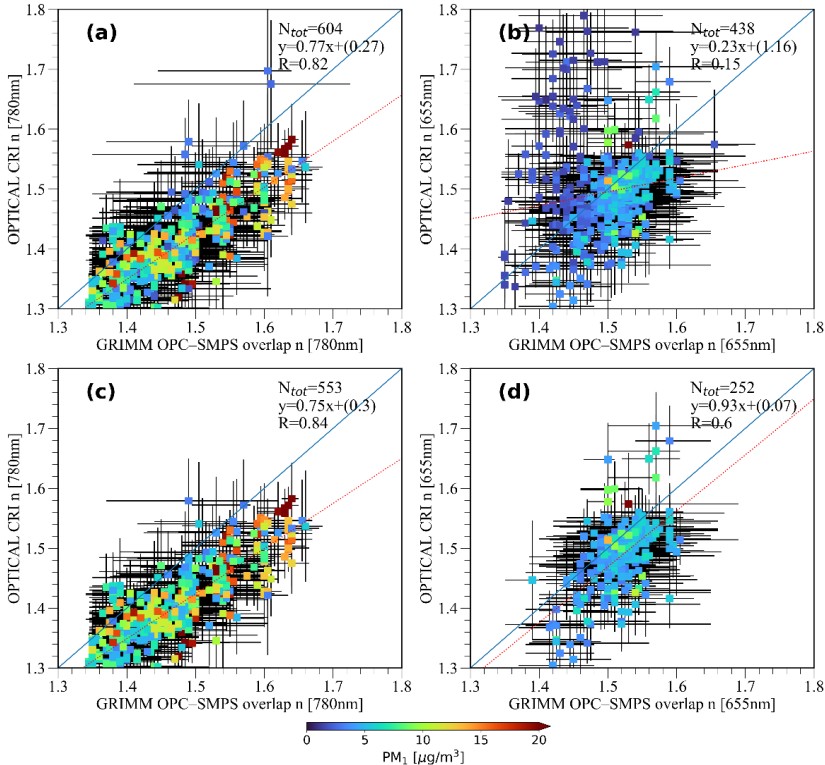


**Figure 2: Scatter plot of the real part of the complex refractive index (CRI) retrieved by applying the optical−closure method (OPTICAL CRI) versus the one retrieved with the OPC−SMPS overlap method at PRG (780 nm, panels a) and c) and RambForest (655 nm, panels b) and d) sites. Points are coloured by the PM1 mass calculated from the SMPS size distribution data assuming an aerosol particle density of 1.4 g cm−3. Panels a) and b) show all data while c) and d) reports data selected using a threshold of PM1>3**
**μg m−3. The red dotted line represents the linear fit (y=ax+b), while the 1:1 line is shown in blue.**



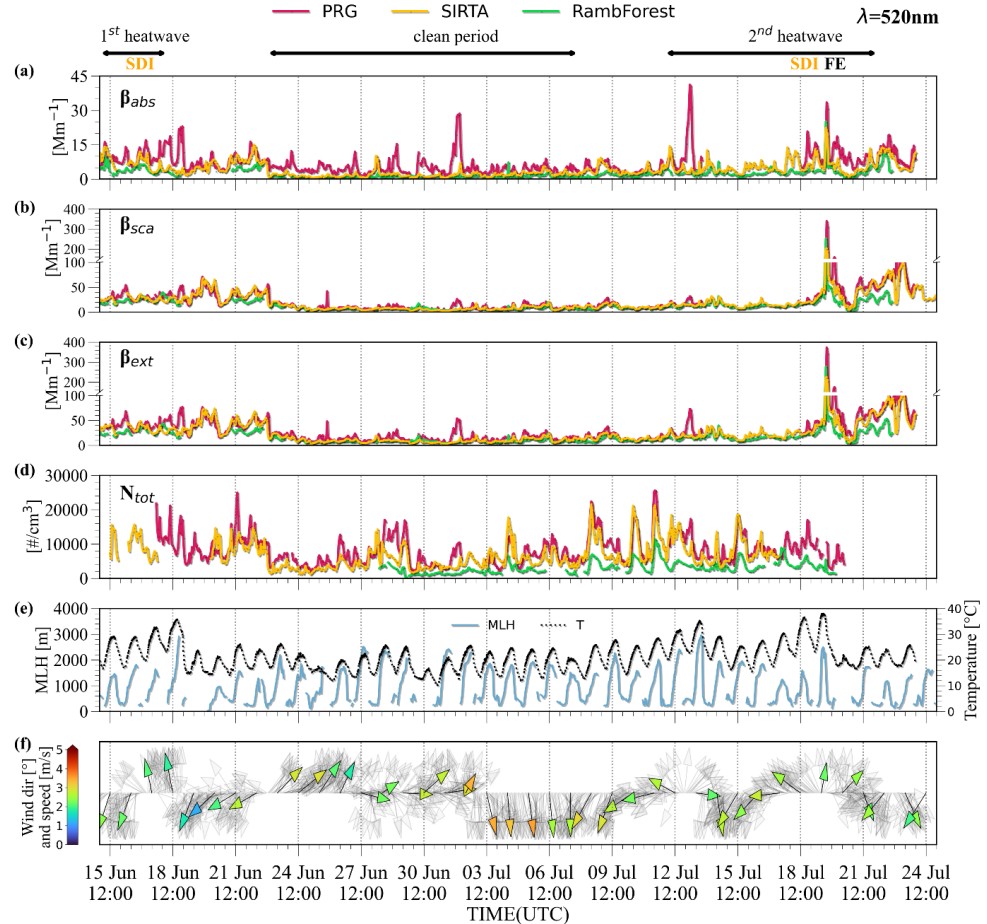

Figure 3: Absorption (βabs), scattering (βsca), and extinction (calculated as βabs+ βsca) coefficients time series at 520 nm at (a) Paris Rive Gauche (urban site) (b) SIRTA (peri–urban site) c) Rambouillet (RambForest) (forest site). Panel (d) represents the total number of particles at the three sites; the panel (e) represents the mixing layer height (MLH) and temperature registered at the SIRTA site. Panel (f) shows the daily wind speed and direction at the SIRTA site. Shaded arrows represent the hourly wind direction. Two different heatwaves periods correlated with the high optical signals during June and July 2022 months, interspersed by a low anthropogenic emission period (defined "clean period") are indicated by arrows at the top of panel a). The labels SDI (coloured in orange) and FE (coloured in black) indicate periods affected by Saharan dust intrusion from the upper levels down to the ground and the long−range transport fire episode which occurred on the 19th of July, respectively. The empty spaces represent periods where measurements were not validated.





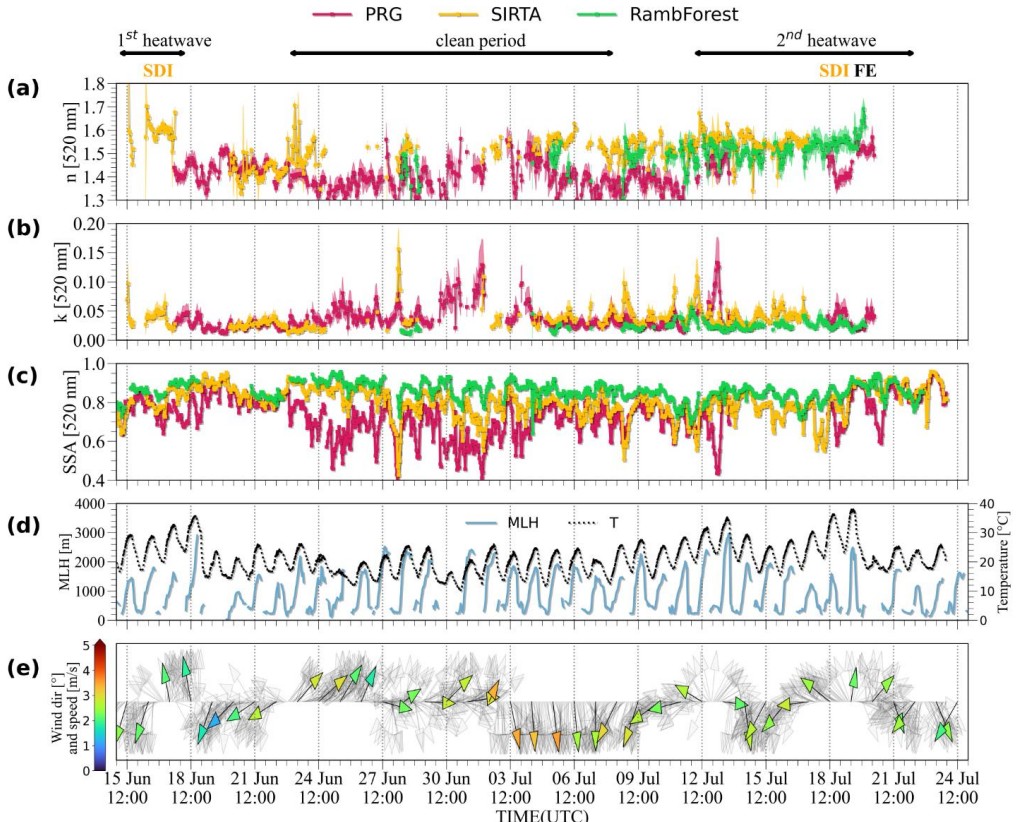

**Figure 4: Time series of the a) real and b) imaginary part of the complex refractive index (CRI) and c) single scattering albedo (SSA at 520 nm retrieved for the ACROSS campaign at the PRG (urban), SIRTA (peri–urban) and RambForest (forest) sites. The mean CRI ± SD is reported. Black arrows at the top of the plots represent the different periods observed during the ACROSS campaign (see main text and 1) first heatwave from 15 June to 18 June 2022; 2) clean period from 23 June to 11 July 2022; 3) the second heatwave from the 12 July to 25 July 2022. The label SDI (coloured in orange) and FE (coloured in black) indicates periods affected by Saharan dust intrusion from the upper levels down to the ground and the long–range transport fire episode occurred on the 19th of July, respectively.**



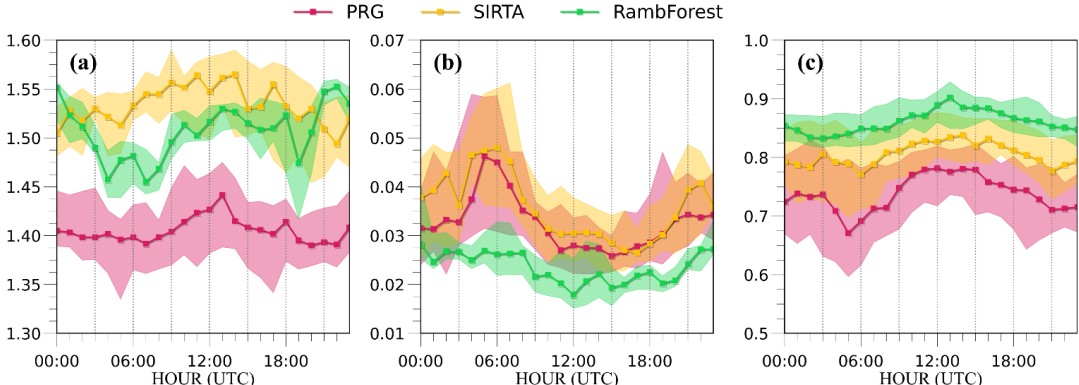

**Figure 5: Diel cycle of hourly values of the (a) real and (b) imaginary part of the complex refractive index (CRI) and (c) single scattering albedo (SSA) at 520 nm retrieved for the full period of the ACROSS campaign at the PRG (urban), SIRTA (peri–urban) and RambForest (forest) sites. The median CRI and SSA are reported. Shaded area represents the 25th and 75th percentiles of the series.**





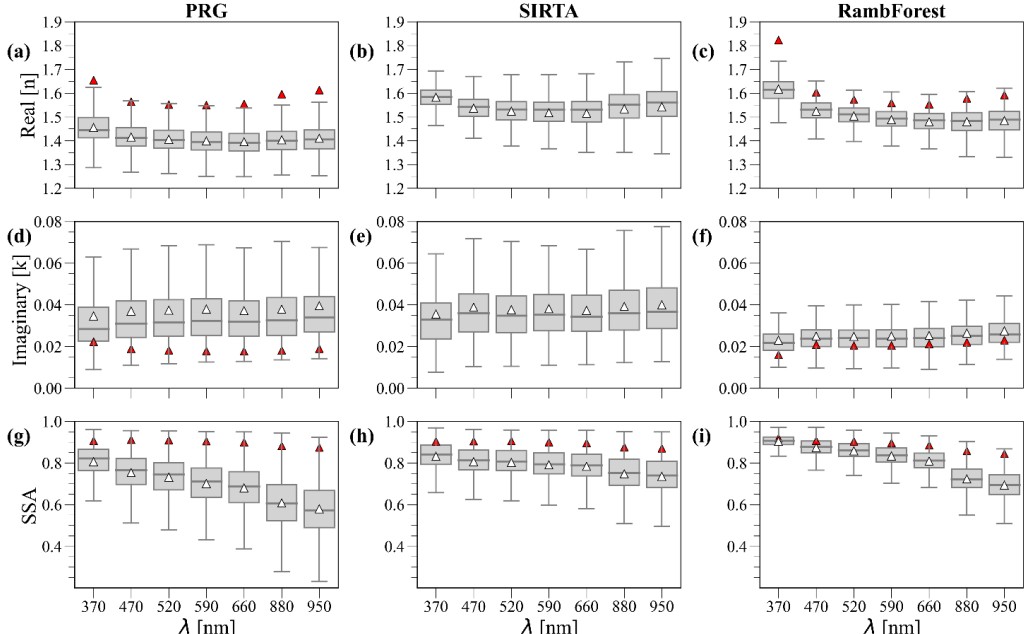

**Figure 6.** Wavelength dependence of the real and imaginary part of the complex refractive index (CRI) and single scattering albedo (SSA) for the full ACROSS period for the PRG (panels (a), (d), (g)), SIRTA (panels (b), (e), (h)) and RambForest (panels (c), (f), (i)) sites. Red triangles show the average values during the fire episode (FE): between 18 UTC and 19 UTC at the urban and peri−urban site, while between 16 UTC and 18 UTC at the forest site). No size distribution data are available for the complex refractive index retrieval at the peri−urban site during the FE. White triangles show the mean value, while black lines represent the medians. Outliers are not shown for the sake of readability.





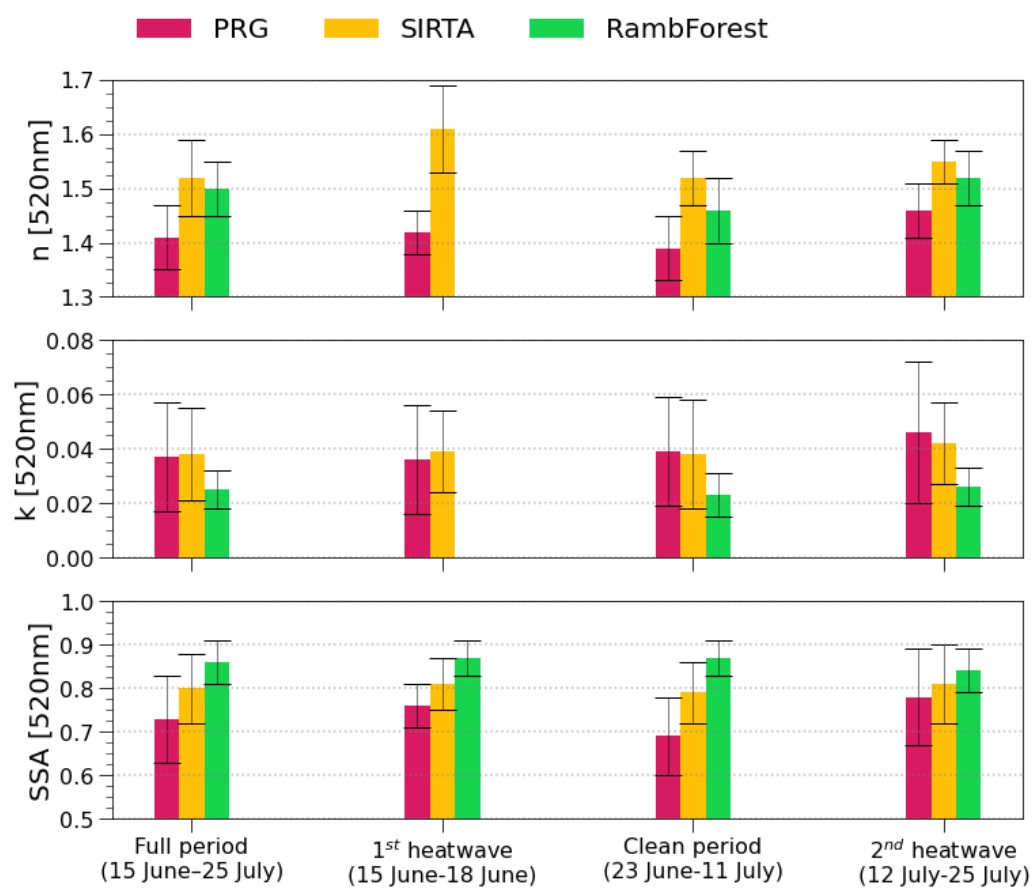

**Figure 7: Average real (n) and complex (k) parts of the CRI and single scattering albedo (SSA) at 520 nm for the full period, the two**
**heatwaves and the clean period for the PRG (urban), SIRTA (peri−urban) and RambForest (forest). Black bars correspond to the**
**SD.**



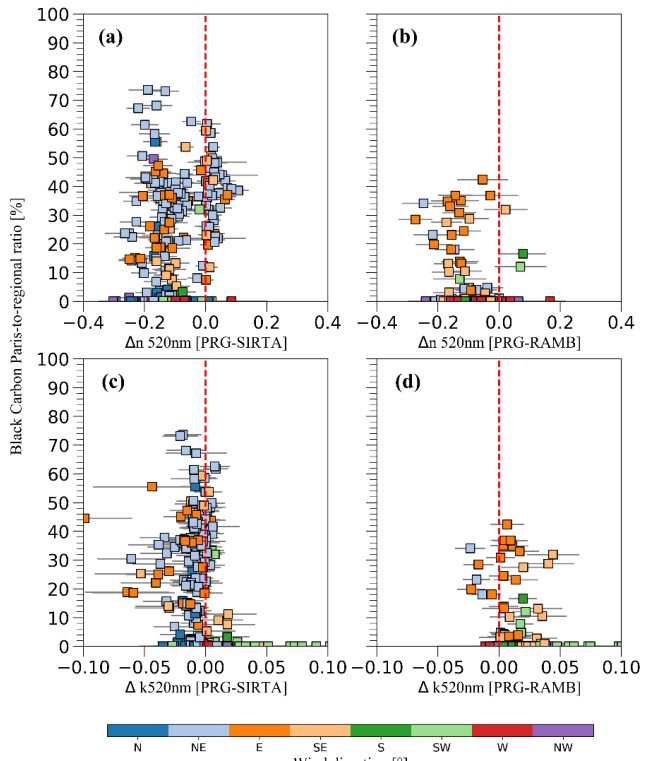

**Figure 8: Scatter plot of absolute differences of the real (Δn, a, b) and imaginary (Δk, c, d) part of the complex refractive index at 470 nm vs the BCratio expressed in %, representing the Grand Paris area BC contribution to the total BC concentration extracted from the CHIMERE model simulation at a spatial resolution of 2 km at the SIRTA (peri–urban, panel (a) and (c)) and the RambForest (forest, panel (b) and (c)) sites. Panels (a) and (c) show the Δn and Δk calculated as difference between PRG and SIRTA data, while (b) and (d) show the Δn and Δk for PRG minus RambForest. The vertical dashed line represents the zero difference line.**

**Horizontal error bars represent ±SD and have been calculated as SD= $\sqrt{SD_1{}^2 + SD_2{}^2}$, where subscripts 1 or 2 stands for the specific site used to perform the difference.**



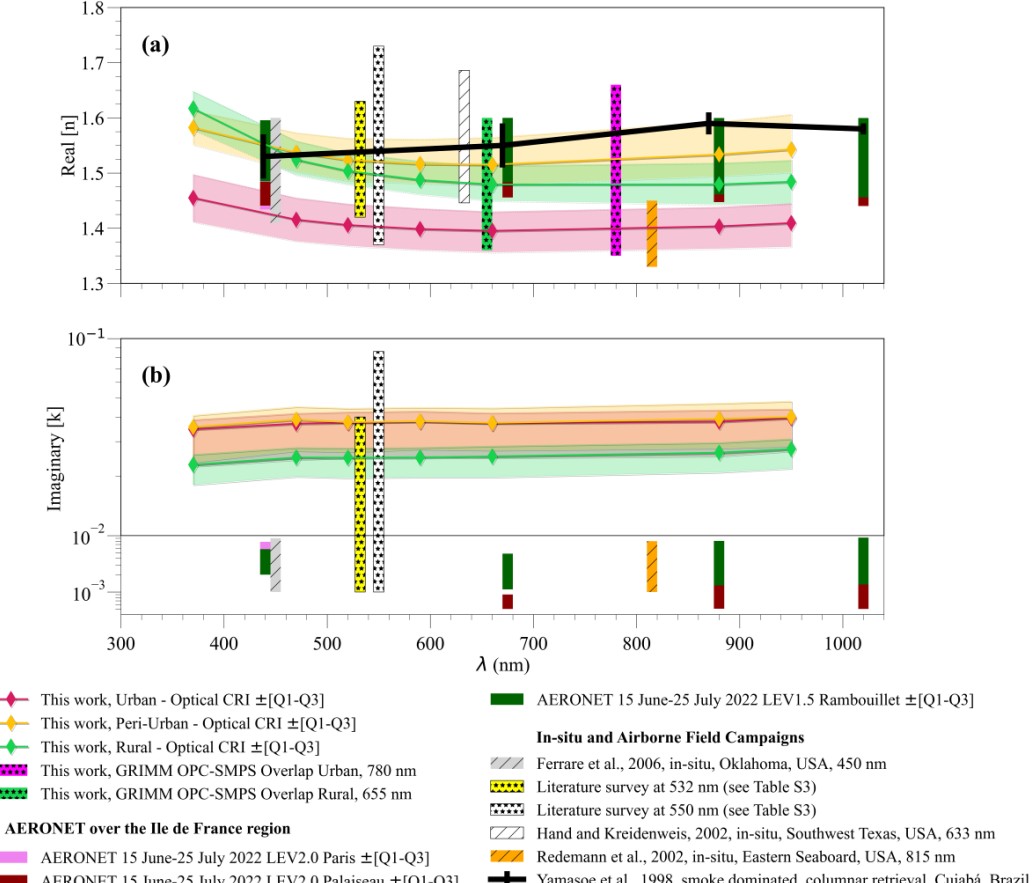

**Figure 9: Comparison of the results obtained in this work with literature surveys of the (a) real (n) and (b) imaginary (k) parts of the complex refractive index (CRI). Mean (solid line), shaded area represents the interquartile range of the CRI at Paris Rive Gauche (PRG, urban), SIRTA (peri–urban) and Rambouillet (RambForest, forest) for the entire ACROSS period 2022 are represented. The legend identifies the line styles used for the literature works. Note that for k values the line for PRG is mainly hidden under that of SIRTA. The literature survey at 532 and 550 nm is detailed in Table S3. Note the logarithmic scale for the imaginary part of the refractive index.**

1250

1255



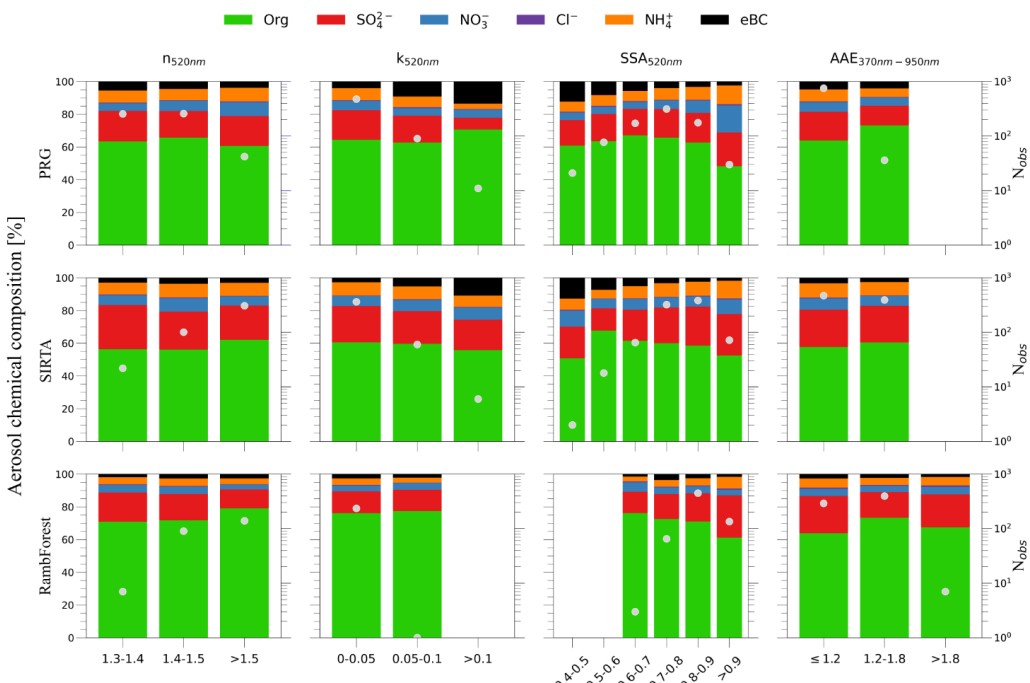

**Figure 10:** Aerosol chemical composition as a function of the complex refractive index (CRI), single scattering albedo (SSA) and absorption Ångstrom exponent (AAE) different classes for the PRG, SIRTA and RambForest sites. White points represent the number of observations in each section. The green colour indicates the organic fraction, the red colour indicates the sulfate fraction, the blue colour indicates the nitrate fraction, violet colour indicates the chloride fraction, the orange colour indicates the ammonium fraction, while black colour indicates the equivalent black carbon contribution.





|  | Full period (15 June–25 July) | | | 1rst heatwave (15 June–18 June) | | | Clean period (23 June–11 July) | | | 2nd Heatwave (12 July–25 July) | | |
|---|---|---|---|---|---|---|---|---|---|---|---|---|
|  | PRG | SIRTA | RambForest | PRG | SIRTA | RambForest | PRG | SIRTA | RambForest | PRG | SIRTA | RambForest |
| $N_{tot}/10^3$ [#/cm3] | 8±4 | 7±4 | 3±2 | 12±4 | 10±3 | – | 7±4 | 6±4 | 3±2 | 9±4 | 8±4 | 4±1 |
| $\beta_{ext}$ [Mm$^{-1}$] | 30±29 | 24±22 | 15±14 | 43±10 | 32±7 | 27±7 | 15±8 | 11±5 | 9±3 | 57±45 | 36±29 | 22±21 |
| $\beta_{sca}$ [Mm$^{-1}$] | 24±26 | 20±20 | 13±13 | 32±7 | 26±5 | 23±6 | 11±6 | 9±4 | 7±3 | 47±42 | 30±26 | 18±18 |
| $\beta_{abs}$ [Mm$^{-1}$] | 7±5 | 4±3 | 2±2 | 10±4 | 6±3 | 4±2 | 5±3 | 2±1 | 1±1 | 10±6 | 5±3 | 3±2 |
| $\beta_{abs}$-BrC [Mm$^{-1}$] | 1.2±1.7 | 1.1±1.1 | 0.9±1.2 | 1.9±0.7 | 1.7±0.9 | 1.5±0.8 | 0.7±0.5 | 0.6±0.4 | 0.5±0.5 | 2.0±3.1 | 1.4±1.5 | 1.1±1.9 |
| AAE | 1.09±0.10 | 1.20±0.08 | 1.21±0.22 | 1.13±0.05 | 1.19±0.06 | 1.29±0.12 | 1.07±0.11 | 1.22±0.08 | 1.19±0.27 | 1.12±0.09 | 1.18±0.09 | 1.22±0.15 |
| SAE | 2.3±0.3 | 1.9±0.4 | 2.7±0.2 | 2.4±0.1 | 2.1±0.4 | 2.7±0.1 | 2.4±0.2 | 1.7±0.4 | 2.8±0.2 | 2.1±0.3 | 1.9±0.2 | 2.7±0.2 |
| n | 1.41±0.06 | 1.52±0.07 | 1.50±0.05 | 1.42±0.04 | 1.61±0.08 | – | 1.39±0.06 | 1.52±0.05 | 1.47±0.06 | 1.46±0.05 | 1.55±0.04 | 1.52±0.05 |
| k | 0.037±0.020 | 0.038±0.017 | 0.025±0.007 | 0.036±0.020 | 0.039±0.015 | – | 0.039±0.020 | 0.038±0.020 | 0.023±0.008 | 0.046±0.026 | 0.042±0.015 | 0.026±0.007 |
| SSA | 0.73±0.10 | 0.80±0.08 | 0.86±0.05 | 0.76±0.05 | 0.81±0.06 | 0.87±0.04 | 0.69±0.09 | 0.79±0.07 | 0.87±0.04 | 0.78±0.11 | 0.81±0.09 | 0.84±0.05 |

**Table 1: Average values ± standard deviation (SD) of the total number of particles (Ntot), extinction (βext) scattering (βsca), and absorption (βabs) coefficients at 520 nm, absorption coefficient fraction due to Brown Carbon (βabs–BrC) at 370 nm, real (n) and imaginary (k) part of CRI and SSA at 520 nm, and absorption and scattering angstrom exponents (AAE, SAE) calculated between 370 and 950 nm for the different periods defined during the ACROSS campaign 2022.**





| Wind direction | SSA ± SD | | | n ± SD | | | k ± SD | | |
|---|---|---|---|---|---|---|---|---|---|
| | PRG | SIRTA | RambForest | PRG | SIRTA | RambForest | PRG | SIRTA | RambForest |
| N | 0.73±0.08 | 0.79±0.06 | 0.86±0.03 | 1.40±0.07 | 1.56±0.05 | 1.48±0.04 | 0.035±0.011 | 0.039±0.010 | 0.021±0.004 |
| NE | 0.77±0.08 | 0.79±0.07 | 0.84±0.04 | 1.40±0.05 | 1.51±0.07 | 1.49±0.05 | 0.028±0.011 | 0.039±0.013 | 0.026±0.006 |
| E | 0.73±0.10 | 0.73±0.11 | 0.80±0.05 | 1.38±0.06 | 1.50±0.06 | 1.50±0.05 | 0.031±0.015 | 0.051±0.031 | 0.032±0.008 |
| SE | 0.72±0.10 | 0.77±0.10 | 0.85±0.05 | 1.42±0.04 | 1.53±0.07 | 1.53±0.04 | 0.043±0.024 | 0.053±0.024 | 0.028±0.007 |
| S | 0.70±0.08 | 0.84±0.05 | 0.90±0.02 | 1.42±0.06 | 1.56±0.04 | 1.51±0.08 | 0.043±0.013 | 0.031±0.014 | 0.018±0.005 |
| SW | 0.67±0.10 | 0.84±0.05 | 0.90±0.03 | 1.39±0.05 | 1.53±0.06 | 1.50±0.05 | 0.048±0.024 | 0.024±0.009 | 0.019±0.004 |
| W | 0.73±0.12 | 0.83±0.06 | 0.87±0.04 | 1.44±0.06 | 1.50±0.07 | 1.54±0.10 | 0.040±0.024 | 0.024±0.006 | 0.021±0.006 |
| NW | 0.77±0.11 | 0.82±0.08 | 0.87±0.04 | 1.41±0.07 | 1.55±0.05 | 1.49±0.06 | 0.034±0.016 | 0.032±0.009 | 0.021±0.006 |

**Table 2: Single scattering albedo (SSA) and real (n) and imaginary (k) parts of the complex refractive index expressed as averages ± standard deviation (SD) at 520 nm as retrieved for the urban (PRG), peri−urban (SIRTA) and forest (RambForest) sites for the different main wind sectors. A more detailed version is available in the supplementary material (Table S4).**



**Appendix A: Useful list of abbreviations and symbols**

AAE Absorption Ångstrom Exponent

ACROSS Atmospheric ChemistRy Of the Suburban foreSt

ACTRIS Aerosol, Clouds and Trace Gases Research Infrastructure

ASOA anthropogenic secondary organic aerosol

AVOC anthropogenic volatile organic compound

BC Black carbon

BrC Brown Carbon

$\beta_{abs-BrC}$ Brown carbon absorption coefficient

BSOA biogenic secondary organic aerosol

BVOC biogenic volatile organic compound

$C_{ref}$ multiple–scattering coefficient

Cref Multiple scattering coefficient

CRI complex refractive index

$C_{sca}$ truncation correction coefficient

$D_g$ geometrical diameter

$D_m$ mobility diameter

$dN(D_g)/dlogD_g$ aerosol number size distribution using geometric diameter $D_g$

$dN(D_m)/dlogD_m$ aerosol number size distribution using mobility diameter $D_m$

$dN(D_{opt})/dlogD_{opt}$ aerosol number size distribution using optical diameter $D_o$

$dN(D_p)/dlogD_p$ aerosol number size distribution

$D_{opt}$ optical diameter

$D_p$ particle diameter

eBC Equivalent black carbon

ERF Effective radiative forcing

ERFaci Effective radiative forcing due to aerosol–cloud interactions

ERFari Effective radiative forcing due to aerosol–radiation interactions

ESQUIF Etude et Simulation de la QUalité de l'air en Ile−de−France

f(x) function of x

FE Fire episode

FF filter factor

H* harmonisation factor

LISAIR Lidar pour la Surveillance de l'AIR

MAE Mean Absolute Error

MLH Mixing layer height

NCEP National Center for Environmental Prediction

$N_{tot}$ total number of particles

OC Organic aerosol

PM Particulate Matter

PRG Paris Rive Gauche

RambForest Rambouillet Forest



RMSD Root Mean Square Difference

RMSE Root Mean Square Error

SAE Scattering Ångstrom Exponent

SAE Scattering Ångström exponent

SSA Single scattering albedo

VOC Volatile Organic Compound

$\beta_{abs}$ aerosol absorption coefficient

$\beta_{atn}$ light attenuation coefficient

$\beta_{sca}$ aerosol scattering coefficient

$\chi$ shape factor