# Peer review of "Aerosol spectral optical properties in the Paris urban area, and its peri-urban and forested surroundings during summer 2022 from ACROSS surface observations"

_EGUsphere, 2024_

## Author Comment (AC1)

At first, we would like to thank the reviewers for having carefully read the paper and providing valuable comments that helped to improve the quality of the manuscript. We have taken into consideration all the comments raised by the reviewers, and have changed the paper accordingly. The details of our changes are highlighted in the text. The point-by-point answers to Reviewers #1 and #2 are provided below.

**Reviewer #1**

**Specific comments:**

1. The CRI retrieval method assumes homogeneous spherical particles. How valid is this assumption for the aerosol in this region, particularly at the urban site which likely has more fresh emissions? Some discussion of the potential impacts of particle morphology/mixing state on the retrievals would be helpful.

We thank the reviewer for this comment. The spherical particle approximation has been demonstrated to be an effective approach in the real atmosphere with a high degree of adoption within the field of chemical transport models and satellite retrievals. This approximation is assumed in the real atmosphere as the level of detail required to model the soot fractal scattering and absorption behavior would require an accurate description of the particle components (e. g. soot morphological properties, including orientation and voids description), which would introduce considerable uncertainty in the optical calculations. Nevertheless, despite the occurrence of fresh emissions in urban environments, the ageing processes leading to particle compacting and coating are likely to occur at a relatively rapid rate under polluted conditions, as more available precursors lead to the formation of coating material (Zhang et al. 2018). This suggests that, although fresh soot particles are initially emitted and may have a fractal shape that deviates from the Mie assumption, they can be conceptualized as particles with a relative spherical equivalent volume. Additionally, given that the measurements are based on a submicron aerosol sampling line, we assume that the contribution of other non-spherical particles, such as coarse dust, is negligible. Finally, the choice of the spherical homogeneous approximation is based on the necessity for consistency with applications such as satellite retrievals and regional/global models where this approximation is often taken into account.
In order to clarify this point, the following comment has been added in the main text (Section. 3.1): "The assumption of particle sphericity, which is also considered for the SMPS diameter correction (Sec. 2.2), is evaluated to reasonably well represent submicron particles, including BC-containing aerosols for which ageing and coating rapidly occur in the urban atmosphere (e.g., Zhang et al., 2018a). The spherical assumption is also in line with aerosol representation in climate models and satellite retrievals, where Mie theory is applied to estimate aerosol optical properties".

Zhang, Y., Zhang, Q., Cheng, Y., Su, H., Li, H., Li, M., Zhang, X., Ding, A., and He, K.: Amplification of light absorption of black carbon associated with air pollution, Atmos. Chem. Phys., 18, 9879–9896, https://doi.org/10.5194/acp-18-9879-2018, 2018.

2.    Can the authors provide any additional context on the size/intensity of the fire plume event on July 19 and typical transport times to the measurement sites? This might help in interpreting the optical property changes observed. Additionally, maybe show aerosol transportation to and from the sites during biomass burning episode and heatwaves in Figure 1 using arrows.

We thank the reviewer for this suggestion. The fire occurred outside the area shown in Figure 1, within the Gironde region of southern France. In order to better describe the fire activity during this period and the transport that occurred on the Ile-de-France (IDF) region, a figure has been added to the supplementary material (Fig S29) including:
- a true colour MODIS satellite image from the MOD021KM product for its passage over France on July 19, 2022 at 10:00 UTC;
- the 8-Day Fire product dataset showing the active fire between July 12 and July 19 using the MODIS Terra Thermal Anomalies/Fire 8-Day L3 product (MOD14A2);
- the 24h WRF-HYSPLIT backtrajectory initialized with the WRF model simulation as detailed in (Siour and Di Antonio, 2023) showing how the air mass has reached the IDF region on July 19 2022.

Furthermore, the following sentence has been added to the text:

"Figure S5 illustrates the long−range transport fire event, integrating the Moderate Resolution Imaging Spectroradiometer (MODIS) satellite data for the July 19 2022 and the HYSPLIT backtrajectories, showing the extent of the fire plume during the second heatwave. It took almost a day for the air masses to rise from southern France to the Paris region."

Siour, G. & Di Antonio, L. (2023). ACROSS_LISA_WRF-CHIMERE_HYSPLIT_Backtraj_1H. [dataset]. Aeris. https://doi.org/10.25326/543

3.    What do you see as the main novel contributions of this study? Specifically, how does the analysis in the Paris region advance our understanding of atmospheric aerosols as compared to previous field studies at other locations?

Current knowledge of the aerosol complex refractive index (real and imaginary parts) from surface-level observations in the real atmosphere, such as urban and rural forested environments, is limited. As a novel contribution, this study presents a comprehensive characterization of the aerosol spectral optical properties under the highly diversified and challenging conditions of summer 2022 within Europe (which include strong and long heatwave periods), which may serve as a proxy for future climate scenarios conditions. The study includes ground-based observations from three sites within the Ile-de-France region, which can be considered as representative of other megacities in Europe and worldwide. A novel contribution from this work is also the analysis of the rural-urban gradient in aerosol optical properties, which shows how optical properties depend on weather conditions and, by integrating modelling studies, also provides evidence of the impact of large urban agglomerations on surrounding sites. Finally, these measurements can provide an added value for the scientific community, such as for satellite retrievals and regional modelling, where there is often limited information on the near-surface aerosol distribution and spatial distribution of their optical properties. These points, which illustrate the novelty of the present study, are highlighted in the conclusion section of the manuscript.

4.    Please consider the following suggestions in these sections:

a. Methods:

i. Consider adding a new section "2.4 Data Processing" to provide step by step details of the data processing, including the quality control and data filtering procedures applied to the raw measurements before and after performing hourly averaging, to obtain the final data shown in the plots.

We thank the reviewer for this suggestion. We believe that the reader can refer to the Supplementary Text 1 and 2 for a detailed description of the CRI retrieval steps. More details on the data treatment applied from the raw data to LEVEL 2 can additionally be found in the abstract of each instrument at https://across.aeris-data.fr/catalogue/ and will be subject of a forthcoming paper to be submitted to Earth System Science Data journal.

ii. Please consider uploading the entire the code script used in the study to GitHub or Zenodo and attach its link/DOI in the code availability statement.

Thanks for the suggestion. The code for the aerosol complex optical retrieval has been added to Github and Zenodo with the following DOI: https://doi.org/10.5281/zenodo.14014328. The "Code availability" section has been modified accordingly.

b. Conclusions:

i. The conclusions could be strengthened by more explicitly stating the implications of these results for aerosol radiative effects and air quality in the region. What are the key takeaways for modelers or policymakers?

We thank the reviewer for this suggestion. The following sentence has been added to the conclusion:
"As a final remark, the large surface spatial and temporal variability of the aerosol optical and chemical properties found within this study, suggests that the aerosol contribution to the aerosol direct and indirect radiative effects and air quality can be highly heterogeneous in the surrounding of urban areas, depending on atmospheric conditions and possible transport of pollutants and mixing with emissions from surrounding environments. While BC remains the dominant aerosol absorber within urban areas, it represents only a few percent of the submicron aerosol mass. More than 50% of the mass at the three sites under investigation is due to organic aerosol, suggesting that this species is an additional major contributor to both air quality degradation and absorption of atmospheric radiation in a broader area around urban agglomerations. To resolve this identified heterogeneity in aerosol distribution and properties, regional chemistry and transport models with high spatial resolution and detailed representation of brown carbon should be adopted to better resolve the temporal and spatial variability of the aerosol load and radiative effects.

Due to the highly diversified and challenging atmospheric conditions observed during summer 2022, proxy of a possible future climate scenario, the measurements presented in this study can serve to the scientific community not only to evaluate the spectral optical properties used to drive the aerosol radiative calculations, but also to assess aerosol satellite retrievals in complex and mixed environmental scenes."

ii.  Consider adding a brief paragraph on recommended potential future work which can build upon this study.

We thank the reviewer for this suggestion. The need for more detailed measurements of the aerosol mixing state and chemical composition is emphasized in the conclusions (lines 552-557). These measurements are needed to better resolve the evolution of the aerosol optical properties presented in this study and to link them to particle chemistry and ageing. Besides, the reviewer can refer to the previous comment for additional future possible applications.

**Technical corrections:**

Line 133: "a.g.l." should be defined on first use

Thanks. The text has been modified as "30 m a.g.l. (above ground level)" and "a.g.l." has been added to the Appendix A section.

Eqn. 5 & 6: Missing space between "370" and "nm"

Thanks, corrected.

Table 2 caption: "averages" should be "average"

Thanks, corrected.

Throughout: Capitalization of "black carbon" as "Black Carbon"

We thank the reviewer for this suggestion but we prefer to keep "black carbon" in lowercase letters throughout the text.

Figure 5: Missing n, k, and SSA text on the y-axis in the figures.

We thank the reviewer for this suggestion. We have corrected the Figure 5.

**Reviewer #2**

Page 3, line 112: The RambForest site is described as being "downwind" of Paris. However, the later analysis shows that this is often not true, depending on the wind conditions, and that it is often upwind of the city, or neither. I would address this here, to set up the reader to understand that there are various wind patterns that lead to urban outflow vs more stagnant air.

We thank the reviewer for the suggestion and we hav removed the word "downwind" word from line 112 as it can be confusing. In addition, we added the following sentence to clarify this aspect: "The surrounding sites are located both upwind and downwind of Paris, depending on the wind direction, allowing for the investigation of different aerosol levels and conditions, from background to polluted."

Page 6, line 221: The dynamic shape factor can be different than 1, particularly when the aerosol contains black carbon. Do you expect that accounting for this could close the gap between the results of the optical-iterative and the OPC-SMPS methods?

We thank the reviewer for this comment. The reviewer can refer to the specific comment #1 from the Reviewer #1 describing why we adopted the spherical particle approximation. Under polluted conditions, such as in urban areas, particle sphericity can be quickly assumed for black carbon due to more available coating material (Zhang et al. 2018). Therefore, we do not expect the shape factor of ambient black carbon shape factor to significantly deviate from unity and to be able to resolve the gap in the OPC-SMPS retrieval.

Zhang, Y., Zhang, Q., Cheng, Y., Su, H., Li, H., Li, M., Zhang, X., Ding, A., and He, K.: Amplification of light absorption of black carbon associated with air pollution, Atmos. Chem. Phys., 18, 9879–9896, https://doi.org/10.5194/acp-18-9879-2018, 2018.

Page 10, line 379: It's not clear whether the Saharan dust intrusion and the fire episode happened at the same time, or slightly offset in time. If the former, how can these two events be separated from each other?

Thanks for this comment. Indeed, two Saharan dust episodes occurred during the ACROSS period, associated with the two heatwaves (the first in June and the second in July 2022). During the second event, on July 19, the fire and Saharan dust episodes occurred simultaneously and discerning between the two is indeed very complex, as correctly indicated by the reviewer, since both can contribute to the UV absorption measured at the ground level. Nevertheless, as our optical calculations are based on submicron input measurements, we expect a reduced contribution from dust compared to biomass burning in the submicron fraction. Additionally, since two Saharian dust intrusions occurred, one concurrently with fire and one without, we were able to evaluate the differences in the spectral absorption coefficient in the two cases. No significant absorption enhancement at 370 nm was measured for the Saharan dust event alone in June 2022, while a measurable absorption enhancement at 370 nm was detected during the Saharan dust + fire episode. Thus, while the contributions of the Saharan dust and fire particles cannot be separated, we assume from observations that the latter more strongly affects the optical aerosol signal at UV wavelengths.

Page 12, line 439: "as the latter one is located NW of Paris". Should this read "SW"?

Yes, the reviewer is correct. Thanks for the correction.

Figure 1: I suggest indicating on the map where the fire occurred.

Thanks for this suggestion. The reviewer can refer to the answer to the specific comment #2 from the Reviewer #1. As the fire event occurred outside the map in Fig. 1, it is not possible to add this information in the Figure. Nonetheless, a supplementary Figure S5 has been added to illustrate the location and the transport pattern of the fire event over the Ile de France region.

Figure 2: The plots reference GRIMM, without indicating what this acronym stands for. I also recommend putting a PRG label over panels a and c and a RambForest label over panels b and d.

Yes, the reviewer is right. Grimm Aerosol Technik is the manufacturer of the optical particle counter (OPC). The figure has been corrected by removing the GRIMM label and adding the PRG and RambForest label.